# Imaging vesicle formation dynamics supports the flexible model of clathrin-mediated endocytosis

Tomasz J. Nawara[1], Yancey D. Williams II [1], Tejeshwar C. Rao[1], Yuesong Hu[2], Elizabeth Sztul[1], Khalid Salaita [2] & Alexa L. Mattheyses [1✉]

Clathrin polymerization and changes in plasma membrane architecture are necessary steps in forming vesicles to internalize cargo during clathrin-mediated endocytosis (CME). Simultaneous analysis of clathrin dynamics and membrane structure is challenging due to the limited axial resolution of fluorescence microscopes and the heterogeneity of CME. This has fueled conflicting models of vesicle assembly and obscured the roles of flat clathrin assemblies. Here, using Simultaneous Two-wavelength Axial Ratiometry (STAR) microscopy, we bridge this critical knowledge gap by quantifying the nanoscale dynamics of clathrin-coat shape change during vesicle assembly. We find that de novo clathrin accumulations generate both flat and curved structures. High-throughput analysis reveals that the initiation of vesicle curvature does not directly correlate with clathrin accumulation. We show clathrin accumulation is preferentially simultaneous with curvature formation at shorter-lived clathrin-coated vesicles (CCVs), but favors a flat-to-curved transition at longer-lived CCVs. The broad spectrum of curvature initiation dynamics revealed by STAR microscopy supports multiple productive mechanisms of vesicle formation and advocates for the flexible model of CME.

[1] Department of Cell, Developmental, and Integrative Biology, University of Alabama at Birmingham, Birmingham, AL, USA. [2] Department of Chemistry, Emory University, Atlanta, GA, USA. ✉email: mattheyes@uab.edu

A tremendous amount of work has explored the mechanism of clathrin-mediated endocytosis (CME) as it plays a critical role in processes including signaling, receptor clearance, morphogenesis, and pathogen internalization. Disruptions in CME are often embryonic lethal or cause severe pathologies[1–7]. Clathrin, a building block of clathrin-coated vesicles (CCVs), is a trimeric coat protein composed of three heavy and three light chains. This coat protein, along with endocytic adaptor and accessory proteins and lipids, mediates cargo recruitment, membrane bending, pit formation, maturation, and scission[7,8]. The coordinated dynamics and interactions of proteins at endocytic sites are well described[9,10]. However, the dynamics of vesicle formation and the role of clathrin and other endocytic proteins in sensing or inducing membrane curvature in living mammalian cells remain under-investigated due to a lack of real-time nanoscale analysis[11–14].

A variety of clathrin-coated structures (CCSs) have been identified at the plasma membrane (PM) by electron microscopy, including flat clathrin lattices (FCLs), domed lattices, clathrin-coated pits (CCPs), and vesicles of different sizes[15,16]. The abundance of different clathrin structures is cell line or tissue-dependent and can be altered by PM tension[17–19]. However, electron microscopy cannot detect curvature formation in real time due to the need for sample fixation. Hence, the fates of different clathrin structures, vesicle formation dynamics, and the proteins driving these processes remains ambiguous. In contrast, live-cell fluorescence microscopy provides dynamic single vesicle resolution, which has contributed to our understanding of protein dynamics and localization at endocytic sites[8,9,13]. However, the limited lateral and axial resolution makes conventional fluorescence microscopy unsuitable for revealing the morphology of clathrin structures during the highly dynamic process of CME[20]. The lack of insight into coat curvature dynamics during CME represents a critical knowledge gap. To circumvent these challenges, data from individual clathrin-coated pits can be sorted based on intensity changes, signal co-localization profiles, or protein lifetimes to make conclusions about morphology, but these approaches are indirect and potentially exclusionary. Consequently, we lack an understanding of the dynamics and roles of different clathrin assemblies and their relative contributions to CME.

This fundamental knowledge gap led to two contrasting models of vesicle formation[21,22]. In the constant curvature model (CCM —Fig. 1ai) curvature is induced simultaneously with clathrin polymerization[23–25]. In the constant area model or flat-to-curved transition (FTC—Fig. 1aii), clathrin assembles as a flat lattice that undergoes a conformational change to form a CCV. Such a rearrangement of clathrin molecules may be energetically unfavorable since, as proposed by some models, FCLs comprise hexagonal clathrin connections, and CCVs are a mix of hexagonal and pentagonal clathrin bonds[24,26,27]. That would place FCLs (Fig. 1aiii) as either frustrated (abortive) sites of endocytosis or in a role outside of CME such as clathrin-containing adhesion complexes, signaling platforms, or stable sites for the cargo recruitment[28–31]. Alternatively, the presence of pentagonal clathrin bonds and structural defects identified within FCLs, could offset the need for clathrin reorganization during vesicle formation[19]. Finally, it has been proposed that there is a flexible balance between these models that incorporates intermediate paths of CCV formation in living cells[12,22,32,33]. The flexible model encompasses CCM and FTC, as well as "in between" pathways. For example, 10% of total clathrin molecules could initially generate a flat structure, following the FTC model, and the remaining 90% of clathrin is recruited along with changes in curvature, following the CCM model. The flexible model remains open for many possible combinations of protein accumulation and membrane shape. To address the mechanism of CCV formation and the role of flat clathrin lattices, cutting edge microscopy approaches including fluorescence polarization[12], correlative super-resolution light and transmission electron microscopy[34], and differential evanescence nanometry[35] have been developed and applied, but the mechanism remains not fully understood[15].

Total internal reflection fluorescence (TIRF) microscopy allows for visualization of fluorescently tagged proteins at or in close proximity (~100–200 nm) to the PM. The exponential decay of the evanescent excitation field has been leveraged to study axial dynamics of exocytic vesicles, as vesicles containing fluorescent cargo become brighter as they approach the PM[36]. Unfortunately, this relationship between intensity and axial position is not sufficient for studying endocytosis, as intensity is inversely impacted by increasing protein density and progression of PM curvature. To overcome this limitation, a TIRF based ratiometric approach called STAR[20] was developed, that takes advantage of the wavelength dependent penetration depth of the evanescent field (Fig. 1b, Eqs. (1) and (2))[37]. By simultaneously imaging two fluorophores with spectrally separated excitation wavelengths (e.g., EGFP, 488 nm; iRFP713, 647 nm) STAR has the unique ability to resolve both protein accumulation (fluorescence intensity) and nanometer z-distribution (fluorescence ratio, Δz) in real time. The changes in intensity ratio between the two fluorophores reports the change in axial distribution of the dual-tagged protein (Eqs. (3) and (4)). Importantly the signal from both fluorophores is collected simultaneously, and the ratio will not be influenced by fast protein dynamics. The axial resolution of STAR microscopy depends on object signal to noise ratio (S/N)[20], with up to 5 nm axial resolution for high S/N objects. Fluorophores closer to the PM are brighter, and contribute to the higher axial resolution of STAR microscopy close to the PM. This makes STAR microscopy an ideal tool for studying the initiation of vesicle curvature formation. STAR microscopy reports the z-distribution of the dual-tagged protein. Given what is known about CME, when we image dual-tagged CLCa the STAR readout reflects PM curvature formation during vesicle assembly. Clathrin accumulation at the PM requires adaptor proteins, such as AP2, that link clathrin and PM[38]. As a vesicle invaginates there will be corresponding changes in the z-distribution of clathrin allowing CLCa-STAR to indirectly report PM shape and curvature formation.

Here we use STAR microscopy to address how de novo, diffraction-limited clathrin-coated vesicles form in living cells. Nanometer axial resolution and millisecond time precision combined with high throughput data analysis allowed us to capture a spectrum clathrin-coat curvature initiation dynamics rather than one dominant pathway, in support of the flexible model of CME. The reported flexibility was universal between epidermal growth factor (EGF) stimulated green monkey kidney fibroblast-like (Cos-7) cells, and vascular endothelial growth factor (VEGF) stimulated primary human umbilical vein endothelial cells (HUVECs). Our results suggest that CME models based on direct correlation of clathrin recruitment and vesicle formation will likely fail to encompass the heterogeneity of CME.

## Results

**STAR microscopy delineates between dynamic flat and curved clathrin accumulations**. STAR microscopy has previously been used to study epidermal growth factor receptor (EGFR) dynamics, and suggested that EGFR clustering sometimes, but not always, corresponds to curvature or corelates with successful endocytosis[20]. To apply STAR to study CME we first validated the method by imaging fluorescently dual-labeled 5 μm microspheres in a refractive index matched solution (Fig. 1c). Widefield

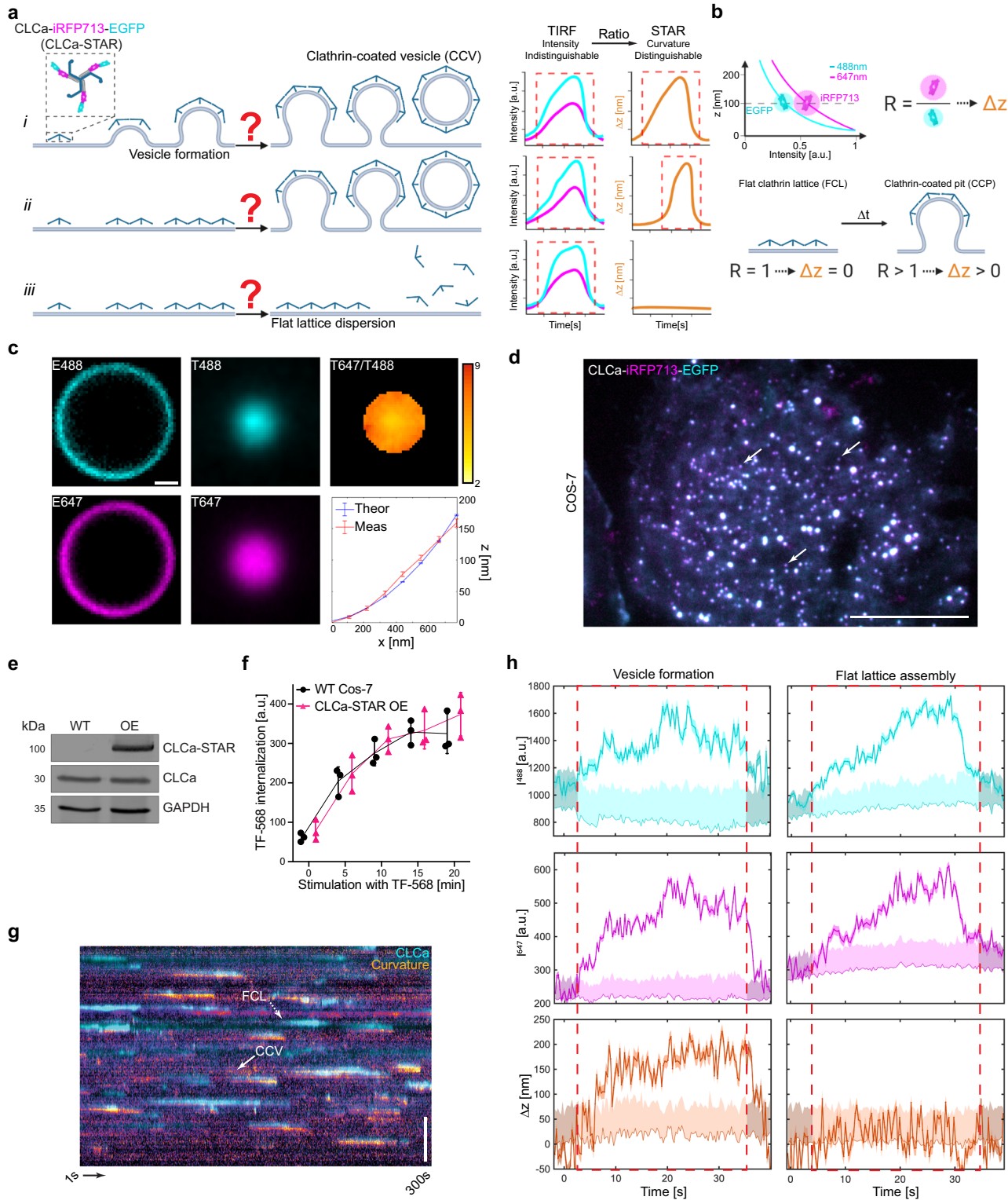

images showed that the labeling was uniform, and the microspheres were monodispersed. TIRF illuminates only the "bottom" of the microsphere in contact with the coverslip, and the larger penetration depth of the 647 nm excitation is reflected by the larger fluorescent zone under the bead. The height profile ($\Delta z$), calculated from the fluorescence intensity ratio of 31 microspheres, was in agreement with the theoretical profile calculated based on the microsphere geometry[20]. This confirms STAR accurately reports changes in axial position ($\Delta z$).

We then used STAR to investigate the dynamics of clathrin assemblies at the PM and determined their contributions to CME. To do so, we dual-tagged the C-terminal of clathrin light chain A with an iRFP713-EGFP fusion (CLCa-STAR). Transient transfection of CLCa-STAR into Cos-7 cells revealed its correct punctate localization at the PM and expression at the predicted molecular weight (Fig. 1d, e). We tested the functionality of CLCa-STAR in a transferrin uptake assay[39–41] comparing wild type and CLCa-STAR expressing Cos-7 cells. No statistically significant

**Fig. 1 STAR microscopy reveals the dynamics of CCV formation in living cells. a** Principles of STAR microscopy. The evanescent field decay is wavelength dependent, and STAR is based on the changing fluorescence ratio of two spectrally separated fluorophores. STAR imaging of CLCa dual-tagged with iRFP713 and EGFP can be used to study the curvature of clathrin-coated structures. **b** Current models describing formation of clathrin-coated vesicles (CCVs): i, Constant Curvature Model (CCM); ii, Flat-to-curved transition (FTC); iii, Flat clathrin lattice (FCL) accumulation and dispersion. **c** 5 μm silica microsphere dual-tagged with Alexa488 and Alexa647, imaged in epifluorescence (E) and TIRF (T), T647/T488 ratio, and theoretical (Theor—blue) and measured (Meas—red) z-position as a function of distance from the center of the bead, mean ± SEM from 31 beads. **d** Cos-7 cell expressing CLCa-iRFP713-EGFP (STAR) probe imaged simultaneously with 488 and 647 TIRF excitation, arrows indicate example clathrin accumulations, scale bar = 20 μm. Representative of 13 cells and five independent repeats. **e** Western-blot analysis of STAR probe overexpression in Cos-7 cells. Representative blot from two independent replicates. **f** Plot of transferrin internalization over time for WT (black) and STAR probe expressing (magenta) Cos-7 cells. Circles and triangles indicate means from three independent replicates at 0, 5, 10, 15, and 20 min for WT Cos-7 ($n = 93, 93, 93, 93, 93$ cells) and CLCa-STAR OE ($n = 89, 91, 93, 86, 92$ cells). Black and magenta lines represent means ± SD. There was no main effect of STAR probe expression on TF-568 uptake by two-way ANOVA [Probe, Stimulation]—[$F_{(1, 4)} = 0.7909$, $p = 0.42$; $F_{(2.132, 8.526)} = 189.5$, $p < 0.0001$]. Šídák's multiple comparisons test, details in Supplementary Table 1. **g** Kymograph showing clathrin accumulation (cyan) and curvature formation (fire), full arrow indicates vesicle formation, dashed arrow indicates flat clathrin lattice assembly. The curvature channel is translated 3 pixels down for better visualization, scale bar = 5 μm. **h** Quantitative GFP (cyan) and iRFP713 (magenta) intensity and Δz (orange) traces for representative CCV and FCL, darker shaded band—SD of detected signal. Lighter cyan, magenta, and orange indicate background mean and 2*SD above background, red dashed boxes indicate when GFP signal is significantly higher than threshold. **a, b** were created with BioRender.com.

differences were observed between matching time points, indicating expression of CLCa-STAR did not have a dominant effect on endocytic rates (Fig. 1f, Supplementary Fig. 1 and Supplementary Table 1). Next, to synchronize ligand stimulated endocytosis and optimize TIRF imaging[42,43], transfected cells were serum starved for at least 30 min followed by stimulation with epidermal growth factor (EGF, 100 ng/ml)[21] immediately prior to imaging. Cells were imaged simultaneously with 488 nm and 647 nm TIRF excitation and the EGFP and iRFP713 emission was split based on wavelength (Supplementary Fig. 2a). Images were processed with the STAR analysis pipeline to retrieve the dynamic pixel-by-pixel Δz (curvature) measurements (Supplementary Fig. 2b). To account for variations in the morphology of the PM, the reported Δz at time $t_i$ was normalized to the average of the first ten frames ($t_1$-$t_{10}$) of the imaging sequence. This approach was previously shown to produce similar results to a time-matched normalization to adjacent PM[20]. Initially, individual clathrin puncta were identified based on EGFP fluorescence, and we looked at whether curvature was induced at each clathrin accumulation. Both curvature positive (CCV) and curvature negative (FCL) clathrin accumulations were observed throughout the PM (Fig. 1g and Supplementary Movie 1). Interestingly, while the intensity traces of curved and flat events were indistinguishable, STAR allowed us to separate these traces into CCVs, those with Δz rising above the set threshold (two standard deviations above background noise[44]), and FCLs, those with no significant change in Δz (Fig. 1h).

**Diffraction-limited CCVs and FCLs form de novo.** To quantify the dynamics of different clathrin structures and their relative contributions to CME, we performed a high throughput screening of 1948 de novo CLCa-STAR accumulations which appeared and disappeared within the imaging sequence from 13 cells using CMEanalysis[44]. The pre-existing clathrin structures and longer-lived (>100 s) clathrin accumulations were excluded from the cohort analysis. EGFP and iRFP713 positive puncta were sorted based on Δz dynamics, revealing populations of CCVs and FCLs at all lifetimes (Fig. 2a, Supplementary Fig. 3). These designations were further confirmed by following the dynamics of individual clathrin accumulations (Fig. 2b), and the examination of their intensity and Δz traces (Fig. 2c). The lifetime distribution of diffraction-limited de novo clathrin accumulations was in agreement with previous reports[44]. We hypothesized that FCLs would predominately be observed in short-lived (<20 s) and/or longer-lived (>50 s) clathrin accumulations, as they could represent frustrated endocytosis or a delayed transition through CME

"checkpoints"[44,45]. Contrary to this hypothesis, we measured FCL formation and dispersion at all lifetimes, with the total number of FCLs lower in populations with longer clathrin lifetimes (Fig. 2d). Nevertheless, there were significantly more curved clathrin accumulations with only ~20% of de novo diffraction-limited clathrin accumulations remaining flat at the PM (Fig. 2e). The ratio of flat-to-curved CCSs in individual cells was not correlated to EGFP intensity at $t_0$, indicating the relative proportions of reported behaviors was not due to CLCa-STAR expression level (Supplementary Fig. 4). Curved and flat events were detected adjacent to one another, indicating that STAR is reporting the dynamics of individual CCSs, and not large-scale PM fluctuations (Fig. 2b). Together, these data show that both CCVs and FCLs are present and have a wide spectrum of lifetimes in living Cos-7 cells upon EGF stimulation.

To ensure our findings with the CLCa-STAR probe reflect endocytic dynamics, we conducted control experiments to test the addition of the iRFP713 fluorophore to the established CLCa-EGFP[10] probe and the impact of its co-factor biliverdin. We compared the localization and dynamics of CLCa-STAR in media supplemented with or without biliverdin to CLCa-EGFP (Supplementary Fig. 5a). Using CMEanalysis, CCSs were identified based on the EGFP signal only. We found no significant difference in the lifetime distribution of endocytic events or the overall event frequency (Supplementary Fig. 5b, c), indicating that dual tagging CLCa and the addition of biliverdin do not disrupt the overall dynamics of clathrin structures at the PM.

To confirm that events with positive Δz changes measured by STAR correspond to vesicle internalization, we performed an epifluorescence (EPI)-STAR analysis. CCVs undergoing internalization should remain visible in EPI following disappearance from TIRF[35,46] and have positive curvature (Δz) changes. In contrast, FCLs should disappear simultaneously from EPI and TIRF with no curvature changes (Supplementary Fig. 6a). Using CMEanalysis, we identified three main groups of events (Supplementary Fig. 6b and Supplementary Movie 2). We found the population of puncta with Δz changes had a delayed disappearance from EPI, while events with no Δz changes disappeared from EPI and TIRF simultaneously. While we observed curved events that disappeared from EPI and TIRF simultaneously, these could represent rapidly internalized, rapidly uncoated CCVs or possibly abortive endocytic events (Supplementary Fig. 6c). These results confirm that STAR microscopy can identify axial dynamics of clathrin puncta and distinguish vesicle formation from flat clathrin assemblies.

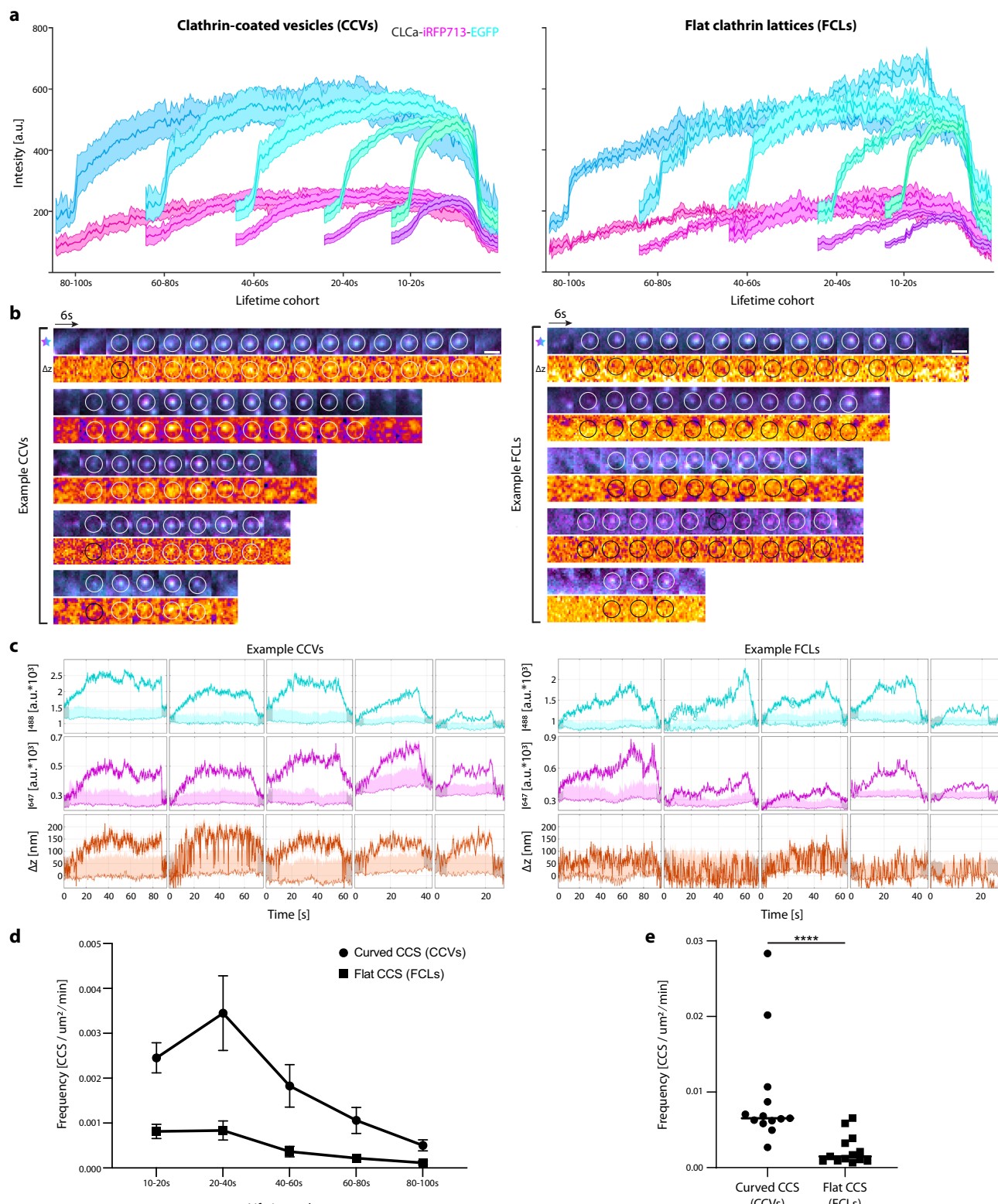

**Spectrum of curvature initiation during CCV formation**. There are three models of how curvature is initiated in CME: (1) Nucleation[47] (Nuc)—PM curvature starts before substantial clathrin accumulation; (2) Constant Curvature[24] (CCM)—PM curvature forms simultaneously with robust clathrin accumulation; (3) Constant area model or flat-to-curved transition[32] (FTC)—PM curvature occurs after full or partial clathrin accumulation, accordingly. We note that since tagged clathrin serves as our

readout, we can only observe nucleation sites where clathrin is present at a low level, possibly prior to its assembly into higher organized structures. The nucleation events reported here represent sites where the EGFP fluorescence intensity is still below the threshold to unambiguously classify the site as a clathrin accumulation site, but the Δz is above the detection threshold sufficient to measure curvature formation. We used STAR to investigate which of these models was driving coat

**Fig. 2 STAR microscopy identifies curved and flat de novo diffraction-limited clathrin accumulations in Cos-7 cells. a** Automated high throughout analysis of live-cell data using CMEanalysis. CLCa-STAR lifetimes in Cos-7 cells separated as positive (left) and negative (right) $\Delta z$, mean ± SEM (EGFP—Cyan, iRFP713—magenta) (CCVs, $n = 1225$ events; FCLs, $n = 328$ events; from 11 cells and three independent experiments). **b** CLCa-STAR intensity (cyan and magenta) and $\Delta z$ (fire) for representative CCVs and FCLs across the lifetime cohorts. Scale bar = 1 μm. **c** Example CCV and FCL signal traces for the events shown in **b**. CLCa-STAR intensity: EGFP (Cyan), iRFP713 (magenta) $\Delta z$ (Orange), darker shaded band represents the SD of detected signal, lighter colors indicate background mean and 2*SD above background. **d** Histogram of lifetime distribution of Curved CCS and Flat CCS per μm$^2$, per minute (mean ± SEM). **e** Cumulative frequency of Curved CCS and Flat CCS per μm$^2$, per minute. Data was not normally distributed as tested by one-sided Shaphiro–Wilk test, black line—median, median for Curved CCS = 0.0065, median for Flat CCS = 0.0015. Medians were significantly different from one another as tested by two-tailed Mann-Whitney test, exact $p < 0.0001$ (Data in **d**, **e**—Curved CCS, $n = 1534$ events, Flat CCS, $n = 414$ events, from 13 cells and five independent repeats).

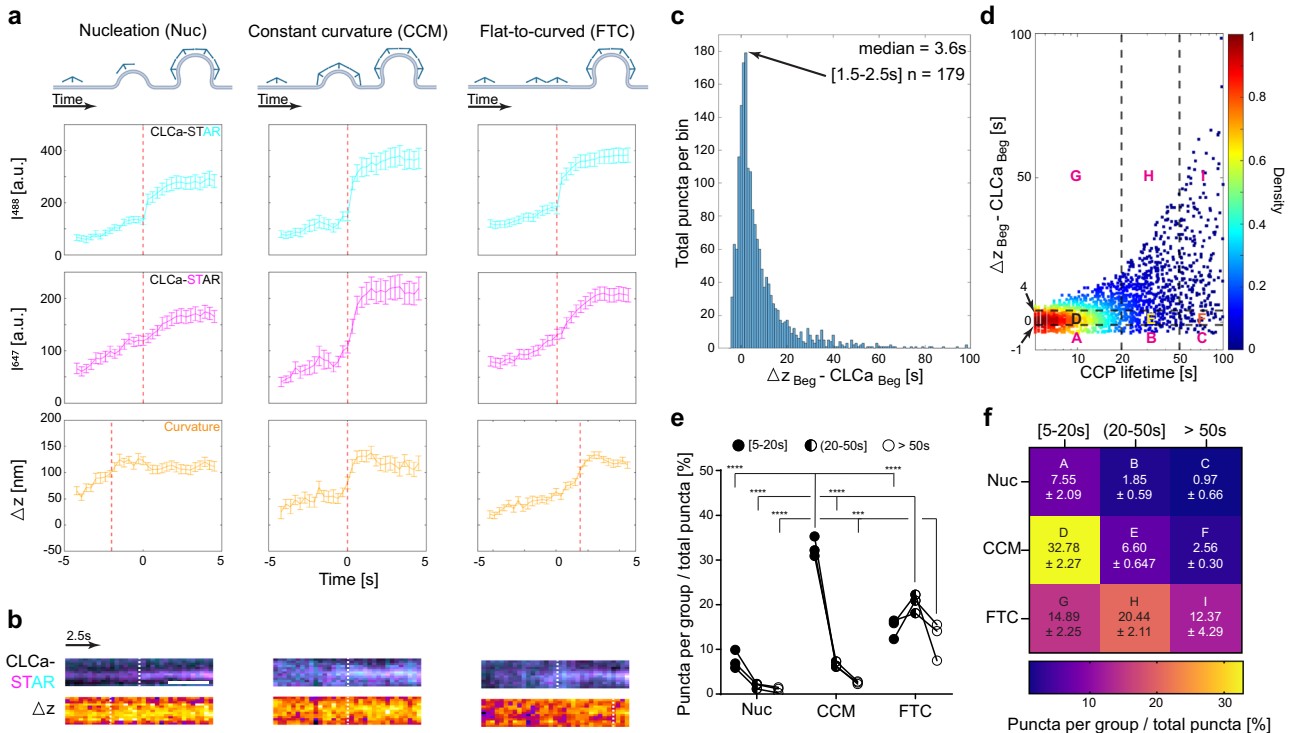

**Fig. 3 CCVs start forming through a gradient of clathrin and curvature lifetimes in Cos-7 cells. a** Models of initial curvature generation during CME and representative intensity and $\Delta z$ measurements, mean ± SEM (Nucleation—$n = 72$ events, Constant curvature—$n = 46$ events, Flat to curved—$n = 144$ events, from 25 Cos-7 cells and three independent repeats). Red dashed line indicates the time where signal reached over the intensity threshold. **b** Kymographs for representative events from **a** (CLCa-STAR, cyan-magenta; $\Delta z$, fire). White dashed line indicates when signal crossed the threshold. Scale bar = 5 μm. **c** Distribution of beginning of curvature formation in living Cos-7 cells measured as a difference between the time when $\Delta z$ reached above threshold and the time when clathrin accumulation reached above threshold ($\Delta z_{Beg} - CLCa_{Beg}$ [s]). **d** Density scatter plot of beginning of curvature ($\Delta z_{Beg} - CLCa_{Beg}$ [s]) as a function of total CCP lifetime (x axis is logged for better visualization, percentages for quadrants A-I are summarized in **f**). **e**, **f** Summary of distribution of events across three models: Nuc = Nucleation ($\Delta z_{Beg} - CLCa_{Beg} < -1s$), CCM = Constant Curvature Model ($1 s \leq \Delta z_{Beg} - CLCa_{Beg} \leq 4$ s), FTC = Flat-to-curved transition ($\Delta z_{Beg} - CLCa_{Beg} > 4$ s). Data from **d** was also classified as events between 5 and 20 s as short-lived, 20–50 s as intermediate, and >50 s as longer-lived. Data presents three independent repeats means and SD are reported as a heat map in **f**. Data was normally distributed (one-sided Shaphiro–Wilk test, $p > 0.05$ for each lifetime cohort), two-way ANOVA was performed (lifetime cohort—$F (2,18) = 93.17$, $p < 0.0001$; PM bending model—$F (2,18) = 93.70$, $p < 0.0001$; interaction—$F (4,18) = 57.49$, $p < 0.0001$). Tukey's multiple comparisons test details can be found in Supplementary Table 2, ***$p = 0.0005$, ****$p < 0.0001$ (Data in **c–f** based on $n = 1805$ events form 25 Cos-7 cells and three independent experiments). **a** was created with BioRender.com.

curvature in all the de novo clathrin accumulations identified by CMEanalysis as single tracks with a lifetime of 5 s or longer and positive for EGFP, iRFP713, and $\Delta z$ (curvature formation). To do so, we defined the beginning of clathrin accumulation ($CLCa_{Beg}$) and curvature formation ($\Delta z_{Beg}$) as the time when the signal of each channel, respectively, crossed above the detection threshold for five consecutive images. We then calculated the difference between $\Delta z_{Beg}$ and $CLCa_{Beg}$, which revealed puncta classified in all three models of curvature initiation (Fig. 3a, b). We defined nucleation as $\Delta z$ (curvature formation) starting <1 s before clathrin accumulation, constant curvature as $\Delta z$ starting 1–4 s after

clathrin arrival, and flat-to-curved transition as $\Delta z$ starting >4 s after clathrin accumulation. The simultaneous detection of clathrin and $\Delta z$ allowed us to capture a spectrum of bending dynamics relative to clathrin accumulations rather than distinct populations, where curvature initiated with a mode of 2 s following clathrin accumulation (Fig. 3c and Supplementary Movie 3).

Since we measured a range of behaviors in when clathrin and $\Delta z$ crossed the threshold, we asked if there was a correlation with the total clathrin lifetime. We hypothesized that the longer persistence of clathrin at PM could indicate delayed initiation of

curvature formation[15]. Surprisingly, we found no obvious correlation (Fig. 3d). To further analyze this data, we grouped endocytic events based on clathrin lifetime and proposed models of vesicle formation. While events were present in all combinations, this revealed some interesting features. We found that short-lived endocytic events (<20 s) were formed predominantly through the CCM, while longer events (>20 s) favored the FTC transition (Fig. 3e, f, and Supplementary Table 2). The proportion of nucleation was smaller and favored short-lived events. It is possible the number of consecutive frames above threshold defined for CLCa and curvature detection could alter our findings and possibly explain the reported spectrum of bending dynamics. We tested this by adjusting the required number of sequential frames above the threshold required to define the beginning of intensity and curvature from 2 to 20 frames (0.6–6 s; Supplementary Fig. 7; Supplementary Table 3). As the signal detection threshold increased, the total number of puncta that fulfill the criteria decreased, as anticipated. However the spectrum of bending dynamics reported at 5 frames (1.5 s) was present throughout and not explained by the detection sensitivity. Together our results provide experimental evidence in favor of the flexible model of endocytosis, where a single model of curvature formation cannot encompass the heterogeneity of CME dynamics.

Our experimental system contained endogenous untagged CLCa and its isoform CLCb[48,49]. The CLC isoforms randomly associate with clathrin triskelia[48,50] and stochastic variation in the recruitment of tagged molecules to CCSs could impact the classification of events based on bending dynamics, especially at the early stages of vesicle formation. At this stage, clathrin intensity is low and a large percent of untagged clathrin at a CCS could result in delayed curvature detection. Alternately, if a majority of tagged clathrin is recruited, the intensity may be detected earlier, also confounding the quantification. To address how stochastic variation in the recruitment of CLCa-STAR and untagged CLCa and CLCb alters $\Delta z$ measurements, we performed a Monte Carlo simulation of STAR measurements during CCV formation. For each stage of vesicle formation, a ground truth center of mass and experimental $\Delta z$ were calculated. The experimental $\Delta z$ was calculated from the average intensity of randomly distributed CLCa-STAR molecules occupying 100–10% of the available points on a Fibonacci sphere (Supplementary Fig. 8a). As expected, the SD of calculated $\Delta z$ measurements was inversely correlated with the percentage of tagged molecules. Nevertheless, we found that STAR measurements followed the ground truth with as little as 30% of tagged molecules contributing to vesicle formation (Fig. 4a, and Supplementary Fig. 8b). The proportion of tagged:untagged CLCa in our experiments is estimated to be at least 50%. This suggests that the stochastic recruitment of tagged and untagged clathrin is not the driving factor underlying the variability reported here.

To experimentally test if recruitment dynamics of wild type and tagged CLCa influenced the lifetime distribution of events, their ability to undergo curvature, and detection of bending models reported by STAR microscopy, we used siRNA to knock down endogenous CLCa followed by expression of CLCa-STAR[29,51,52]. We confirmed CLCa silencing and CLCa-STAR overexpression by western blot (Fig. 4b). We observed correct clathrin localization at the PM in both control siRNA and CLCa targeting siRNA transfected Cos-7 cells, and both curved and flat events were observed throughout the imaging time-lapse (Fig. 4c). Moreover, there were no significant differences in the distribution of events between matching lifetime cohorts (Fig. 4d) and the overall frequency of curved and flat events per cell (Fig. 4e). Since endogenous clathrin expression was minimal, we expected to remove detection artifacts based on stochastic recruitment from

the analysis of curvature initiation. We were particularly interested to investigate whether the presence of the untagged CLCa could explain events that fall into the nucleation category, as untagged CLCa could be driving curvature without a significant fluorescence signal. In both control and CLCa targeting siRNA treated cells we observed a spectrum of bending dynamics (Fig. 4f) and no correlation of curvature formation with CCS lifetime (Fig. 4g). Interestingly, events were distributed across all three bending models with or without the endogenous CLCa, with minor non-statistically significant differences (Fig. 4h, i, and Supplementary Tables 4, 6). This data lets us conclude that the presence of endogenous CLCa in our system (Figs. 2 and 3) did not significantly alter CCSs formation dynamics, the ability to undergo curvature, or the classification of the events within the bending model.

**Flexible model of CME in HUVECs.** To test whether the flexible model of CME is universal between different cell lines, we next imaged vascular endothelial growth factor (VEGF) stimulated HUVECs with STAR. We used VEGF as the ligand to allow comparisons with our findings in Cos-7 cells, as VEGF is one of the key growth factors in this cell line[53]. Following transfection of HUVECs with CLCa-STAR, we confirmed the correct localization of the construct in puncta at the PM and expression at the expected molecular weight (Fig. 5a, b). We found the intensity traces of curved and flat events were indistinguishable, and not every clathrin accumulation resulted in the formation of a CCV (Fig. 5c–e). There were CCVs and FCLs present at all lifetimes, but fewer long-lasting FCLs compared to Cos 7 cells which may represent cell type[17] or ligand/receptor[54] selectivity (Fig. 5f). Overall, there were significantly more curved than flat events, similar to in Cos-7 cells (Fig. 5g). When we analyzed the model of vesicle formation, we found that in the majority of events curvature ($\Delta z$) and clathrin accumulation began simultaneously (Fig. 5h). This suggests the predominant role of the CCM in vesicle formation. As in Cos-7 cells, there was not a correlation between curvature initiation and the total lifetime of clathrin at the PM. Following classification based on lifetime and curvature model, we found that vesicle formation occurred predominantly through the CCM at all lifetimes. While the proportion of FTC events increased at longer lifetimes, the difference was not significant when compared to the CCM events (Fig. 5j, k, and Supplementary Table 7). These data show the previously unappreciated heterogeneity of CME dynamics and plasticity of CCV formation and suggest different cell types or cargos can use this flexibility to impact the mode of curvature formation.

## Discussion

Overall, our results provide evidence in favor of the flexible model[22,33] of CCV formation (Fig. 6). In agreement with previous research, we found the beginning of coat curvature can occur following the initiation of clathrin accumulation[12,55]. Additionally, we identified a population of endocytic events developing curvature prior to detectable accumulation of clathrin. BAR domain proteins, such as FCHO1/2[13,56], can induce PM curvature through insertion of amphipathic helices or molecular crowding[57]. Hence, curvature could develop prior to clathrin polymerization into higher organized structures at nucleation sites. Our results suggest the likely involvement of accessory proteins in early curvature generation[47,52]. STAR microscopy provides a dynamic context for the high-resolution images of clathrin coats with multiple architectures, compositions, and curvatures, reported by EM. Moreover, EM studies provide important context for our results, and lead us to the inclusive view of CCV formation while rejecting the dualism of opposing

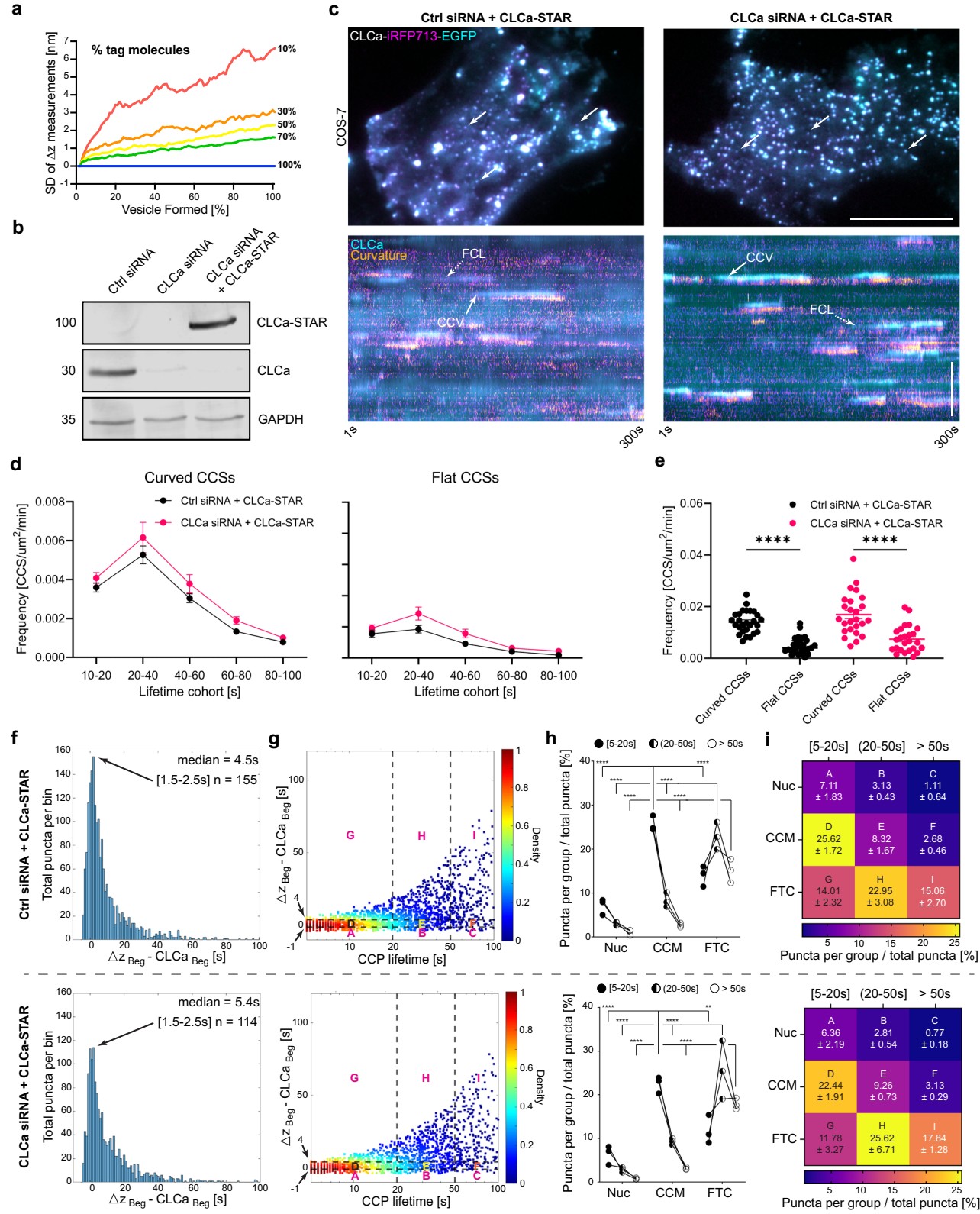

models. Our results put the clathrin structures detected by EM in context of living cells by showing the flexibility of clathrin-curvature interplay, and the involvement of FCLs in CME[15,19,34]. However, we found that not every clathrin accumulation resulted in curvature, and flat lattices had similar intensity dynamics. Our data supports the existence of dynamic flat clathrin lattices that are involved in processes outside of CME such as formation of

adhesion complexes[28], signaling platforms[30], cell spreading and migration[29], dictating cellular fate[9], or tissue differentiation[17]. It is worth noting that we focused on diffraction limited, de novo, and complete CME events, with a total duration shorter than 100 s. While we observed interesting curvature formation dynamics in larger structures, their high throughput analysis is difficult. Lower throughput analysis of such structures may reveal

**Fig. 4 Endogenous CLCa does not explain the variation of curvature formation. a** Standard deviation of STAR measurements from Monte Carlo simulations of vesicle assembly with varying percentages of STAR-probes. **b** Western-blot analysis of CLCa siRNA silencing and STAR probe expression in Cos-7 cells. Representative of two independent replicates. **c** Cos-7 cells treated with control or CLCa targeting siRNA and expressing CLCa-STAR. Arrows —clathrin accumulations, scale bar = 20 μm, and kymograph showing clathrin accumulation (cyan) and curvature formation (fire), full arrow—vesicle formation, dashed arrow—flat clathrin lattice assembly. The curvature channel is translated 3 pixels down, scale bar = 5 μm. **d** Histogram of lifetime distribution of curved and flat CCS per μm², per minute (mean ± SEM). **e** Cumulative frequency of curved CCS and flat CCS per μm², per minute. Black and magenta lines—mean ± SEM, data was normally distributed, except for Ctrl Flat CCSs by one-way Brown-Forsythe ANOVA $F = 27.15$ (3.000, 64.66), exact $p < 0.0001$; medians for: Curved CCSs [Ctrl siRNA; CLCa siRNA] = [0.01362, 0.01498], Flat CCS = [0.003975, 0.003975], ****$p < 0.0001$. $P$ values adjusted for multiple comparison using statistical hypothesis testing. **f** Distribution of beginning of curvature formation measured as $\Delta z_{Beg} - CLCa_{Beg}$ [s] **g** Density scatter plot of $\Delta z_{Beg} - CLCa_{Beg}$ [s] as a function of total CCP lifetime. **h, i** Distribution of events across three models and three lifetime cohorts following the classification form Fig. 3, Nuc = Nucleation, CCM = Constant Curvature Model, FTC = Flat-to-curved transition. Data presents three independent repeats means and SD are reported as a heat map in **i**. Data was normally distributed, two-way ANOVA was performed (lifetime cohort—[$F$ (2, 18) = 54.82, $p < 0.0001$; $F$ (2, 18) = 13.89, $p = 0.0002$]; PM bending model—[$F$ (2, 18) = 118.4, $p < 0.0001$; $F$ (2, 18) = 69.47, $p < 0.0001$]; interaction— [$F$ (2, 18) = 118.4, $p < 0.0001$; $F$ (4, 18) = 24.14, $p < 0.0001$]). Tukey's multiple comparisons test details in Supplementary Tables 4 and 5 and the comparison in Supplementary Table 6, ns = not significant, **$p = 0.0036$, ****$p < 0.0001$. (Data in **d–i**—Curved CCSs $n$ = [2655, 2506] events, Flat CCSs, $n$ = [849, 1048] events, from refs. 27,25 cells and three independent repeats).

even more complex pathways of vesicle formation, such as those identified by atomic force microscopy[58]. Even though we introduced classifications of curvature initiation to allow for quantitative analysis, we do not want to emphasize these boundaries and instead feel the flexibility and spectrum of coat curvature dynamics is central to CME.

Our results suggest that clathrin-centric models, where clathrin recruitment is directly correlated to vesicle formation, will likely fail to encompass the heterogeneity of CME. We speculate that the balance of pathways to initiation of vesicle formation in the flexible model can be shifted by features such as PM tension, cargo size, or local concentration of adaptor proteins[59]. The ability of STAR to measure nanometer axial dynamics will help define the roles of distinct flat and curved clathrin structures. What underlines the variation in PM bending dynamics and whether we can shift the preferred route of the model remains to be investigated. Future research using STAR microscopy will define how the recruitment of accessory proteins and the role of biophysical parameters contribute to vesicle formation and how different clathrin dynamics impact signaling and cellular homeostasis.

## Methods

**Automated data processing.** For automatic detection and tracking of endocytic events using CMEanalysis, TIRF 488, TIRF 647, and Δz channels for each cell were organized as previously published[44] and submitted to the Cheaha supercomputer (UAB IT Research Computing) for processing. The modified CMEanalysis, other MATLAB programs, FIJI scripts, and sample data sets are available on the labs' GitHub and under the code availability section. The executable code and critical parameters are summarized in Supplementary Table 8. Here we analyzed CCSs, identified by CMEanalysis, that formed and left within the imaging time-lapse, which were 5 s or longer, identified as a single track with a valid amount of missing signal (up to 13 frames = 3.9 s; median gap length 1 frame), and positive for EGFP and iRFP713. Tracks identified by CMEanalysis as curvature positive were further processed using custom-written MATLAB programs (Supplementary Fig. 9a and 10a). Briefly, two CMEanalysis outputs, cell masks and detected and processed tracks, are automatically extracted and organized for further processing (Supplementary Figs. 9b and 10b). The distribution of Δz dynamics is automatically quantified and histograms and scatterplots correlating it to event lifetime are created by the MATLAB program cme_wrapper.m. Sorting of curved and flat events and track interpolation was achieved by the MATLAB program cohort_wrapper.m. The step-by-step details to this analysis are presented in Supplementary Figs. 9c and 10c and all the parameters used here are summarized in Supplementary Table 8.

**Cell culture.** Cos-7 cells (ATCC, CRL-1651) were cultured in Dulbecco's modified Eagle's medium (DMEM; Corning, 10013CV) containing L-glutamine and sodium pyruvate and supplemented with 10% fetal bovine serum (Gibco, 10438-026) and 100 IU/ml penicillin-streptomycin (Gibco, 15070-063). Human Umbilical Vein Endothelial Cells (Pooled HUVECs, Lonza, C2519A) were cultured in Endothelial Cell Growth Medium-2 (EGM2, Lonza, CC-3156) supplemented with the BulletKit (Lonza, CC-3162). Cells were maintained at 37 °C and 5% CO₂. Cells were passaged at 90% confluency, resuspended in fresh media, and one-fifth was plated on 100 mm Tissue Cell Culture Dishes (Falcon, 353003). For HUVECs full EGM-2

media was first placed onto a cell culture dish and left to equilibrate into the incubator for 30 min.

**Plasmids.** CLCa tagged with iRFP713 and EGFP (CLCa-iRFP713-EGFP) was constructed and cloned by Oskar Laur at the Emory University cloning core using the Rattus norvegicus CLCa in pEGFP-N1 backbone. The CLCa-iRFP713-EGFP encoding sequence was confirmed by Sanger sequencing at UAB-Heflin sequencing core.

**Microscopy.** Live-cell STAR image acquisition was performed with a Nikon Ti-2 microscope equipped with a motorized stage, stage-top incubator to maintain 37 °C and 5% CO₂ (Tokai Hit, INUBG2SF-TIZB), ×60 1.49-NA objective, manual TIRF illuminator (Nikon, TI-LA-TIRF), 488 nm (Obis, 488-150 LS), and 647 nm (Obis, 1196627) excitation lasers, fiber coupling optics: fiber mount (Thorlabs, MBT621D), converging and directing the laser objective (Olympus, RMS10X), optical fiber (Thorlabs, P3-405BPM-FC-2), C-NSTORM QUAD 405/488/561/ 638 nm TIRF dichroic. Images were acquired with an Optosplit III (Cairn Research) image splitter with ET525/50 m and ET705/72 m emission filters (Chroma), and T562lpxtr-UF2 and T640lpxtr-UF2 dichroic mirrors to split the fluorescence emission onto separate regions of the ORCA-Flash 4.0 v3 scientific complementary metal-oxide-semiconductor camera (Hamamatsu). The system was coupled by a data acquisition device (NIDAQ, National Instruments, BNC-2115) and controlled using Nikon Elements software (version 5.02) and Coherent Connection software (version 3.0.0.8). Image acquisition was performed through NIS JOBS. Optosplit III was calibrated using the manufacturer protocol and the NanoGrid (Miraloma Tech, LLC, A00020).

**Silica microsphere labeling and imaging.** 5 μm silica microspheres (Bangs Labs, SS05003) were labeled as previously described[20]. For imaging, a solution of 58% glycerol in ddH₂0 was prepared to match the refractive index of the beads ($n$ = 1.43). Beads were resuspended in 1 ml of the glycerol solution and placed onto 25 mm #1.5H glass coverslips (Thorlabs, CG15XH1), mounted into the Attofluor cell chamber (Invitrogen, A7816). Beads were imaged in TIRF with sequential 488 and 647 nm excitation, followed by adjustment of the focal plane to the middle of the bead and sequential epifluorescence 488 (Filter cube: 96362 ET GFP C180702) and 647 (Filter cube: 96365 ET CY5 C180340) excitations. All images were taken using 200 ms exposure time. Beads with uneven distribution of fluorophores or misaligned in relation to the epifluorescence signal were discarded from quantification.

**EGFR and VEGFR internalization.** Prior live-cell imaging, 24 h post-transfection, Cos-7 or HUVEC cells grown on 25 mm #1.5H glass coverslips (Thorlabs, CG15XH1) were incubated for at least 30 min in serum-free FluoroBrite DMEM media (Gibco, A1896701) supplemented with 5 μM biliverdin (co-factor for iRFP713; Cayman Chemical, 1925710). Time-lapse TIRF imaging was performed following stimulation with hEGF (100 ng/ml[21]; Millipore Sigma, E9644-.2MG) for Cos-7 cells, or recombinant hVEGF 165 (50 ng/ml[60]; Fischer Scientific, 293VE010CF) for HUVECs. Each cell was imaged simultaneously with TIRF 488 and TIRF 647, for 5 min with 200 ms exposure time and 100 ms delay between frames. Three to four cells were imaged per ligand stimulation.

**Data correction files.** On every experimental day, two types of correction files were generated. (1) Cairn image splitter calibration—stack of 10 images of the NanoGrid (Miraloma Tech, LLC, A00020) with transmitted light illumination. This is required for image splitting and registration. (2) Flat field correction—To correct

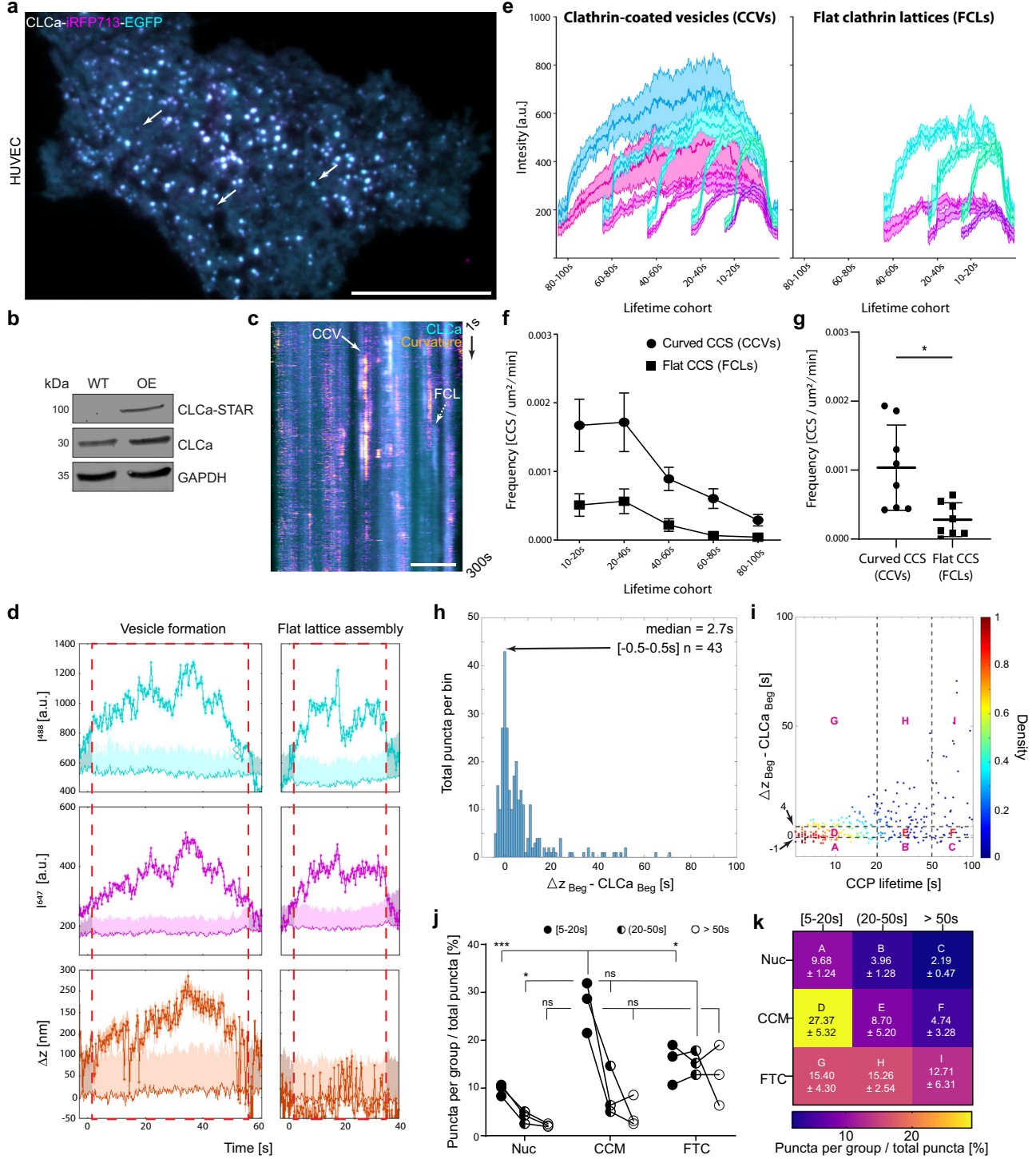

for the inhomogeneity of the TIRF excitation field, a stack of 10 images of fluorescein (Excited by 488 nm laser—ACROS ORGANICS, 2321-07-5) and DiD (Excited by 647 nm laser—Invitrogen, D7757) were acquired. Stock fluorescein was prepared in 1 M NaOH at 1 mg/ml and diluted at 5 µl/ml in NaOH on the day of imaging. DiD was resuspended according to manufacture instructions and diluted at 5 µl/ml in EtOH. The dilutions were applied on clean 25 mm #1.5H glass coverslips (Thorlabs, CG15XH1—coverslips were scratched in middle using a blade to find the focal plane that matches plasma membrane-coverslip interface) and mounted into Attofluor Cell Chambers (Invitrogen, A7816). TIRF images of both coverslips were taken separately with simultaneous 488 and 647 nm excitation to mimic live-cell imaging conditions.

**Generation of Δz (curvature) channel.** A step-by-step data processing pipeline is presented in Supplementary Fig. 2b. To generate the Δz (curvature) channel: First

the 10 images of the NanoGird were stacked based on averaged intensity, and created image was used to generate a calibration file required for the emission signal separation using a Fiji plugin (Cairn Image Splitter version 1.5—https://imagej.nih.gov/ij/plugins/cairn-splitter.html). Post splitting grid pictures were saved as they will be used for image registration. Then, the flat field correction files and experimental data files were split using the calibration file. Flat field correction images were then stacked for each emission signal based on average intensity. Each channel of the experimental data was then background subtracted. Reasonable size (at least 100 × 100 pixels) ROI was drawn outside of the cell area. The mean intensity from the ROI was then subtracted from each channel for every image accordingly. Next, we performed a flat field correction. First, the live-cell data bit depth is changed to 32-bits. The max intensity value is measured from an excitation matching flat field. Then every image was normalized using the respective flat field correction image. Using a built-in Fiji function, we performed a bleach correction (simple ratio method) for both channels separately. The 647 nm channel

**Fig. 5 Majority of CCVs form simultaneously with clathrin arrival in HUVECs. a** HUVEC expressing CLCa-STAR, arrows = clathrin accumulations, scale bar = 20 μm. Representative of 9 HUVECs and three independent repeats. **b** Western-blot analysis of STAR probe expression in HUVECs. Representative blot from three independent replicates. **c** Kymograph showing clathrin accumulation (cyan) and curvature formation (fire), full arrow = vesicle formation, dashed arrow = flat clathrin lattice assembly. Curvature channel is translated 3 pixels to the right, scale bar = 5 μm. **d** Quantitative intensity and Δz traces for CCV and FCL, lighter cyan, magenta and orange = background mean and 2*SD above background, darker shaded band represents the SD of detected signal, red dashed boxes = clathrin signal significantly higher than threshold. **e** CMEanalysis of CLCa-STAR lifetimes separated on whether Δz was generated (mean ± SEM, EGFP—Cyan, iRFP713—magenta). **f** Histogram of lifetime distribution of curved and flat CCS per μm², per minute—mean ± SEM. **g** Cumulative frequency of curved and flat CCS per μm², per minute, mean ± SD, data was normally distributed, two-tailed unpaired *t*-test, $p = 0.0064$, $*p < 0.01$, means: Curved CCS = 0.00104, Flat CCS = 0.00028. (Data in **e–g** based on CCVs—$n = 458$ events, FCLs—$n = 143$ events, from 8 cells and three independent repeats). **h** Distribution of beginning of curvature formation measured as $\Delta z_{Beg} - CLCa_{Beg}$ [s]. **i** Density scatter plot of $\Delta z_{Beg} - CLCa_{Beg}$ [s] as a function of total CCP lifetime. **j**, **k** Distribution across three membrane bending models. Data from **i** was classified as in Fig. 3. Data represents three independent repeats, means and SD reported as a heat map in **k**. Data was normally distributed (one-sided Shaphiro-Wilk test, $p > 0.05$ for each lifetime cohort) and analyzed by two-way ANOVA (lifetime cohort—$F (2,18) = 19.55$, $p < 0.0001$; PM bending model—$F (2,18) = 15.55$, $p = 0.0001$; interaction—$F (4,18) = 6.712$, $p = 0.0017$). Tukey's multiple comparisons test details in Supplementary Table 7, ns = not significant, * (CCM[5–19,20 s] vs FTC[5–19,20 s])—$p = 0.0278$, * (Nuc$_{(20-50s)}$ vs FTC$_{(20-50s)}$)—$p = 0.0423$, $***p = 0.0007$ (Data in **h–k** based on $n = 283$ events form 9 HUVEC cells and three independent repeats).

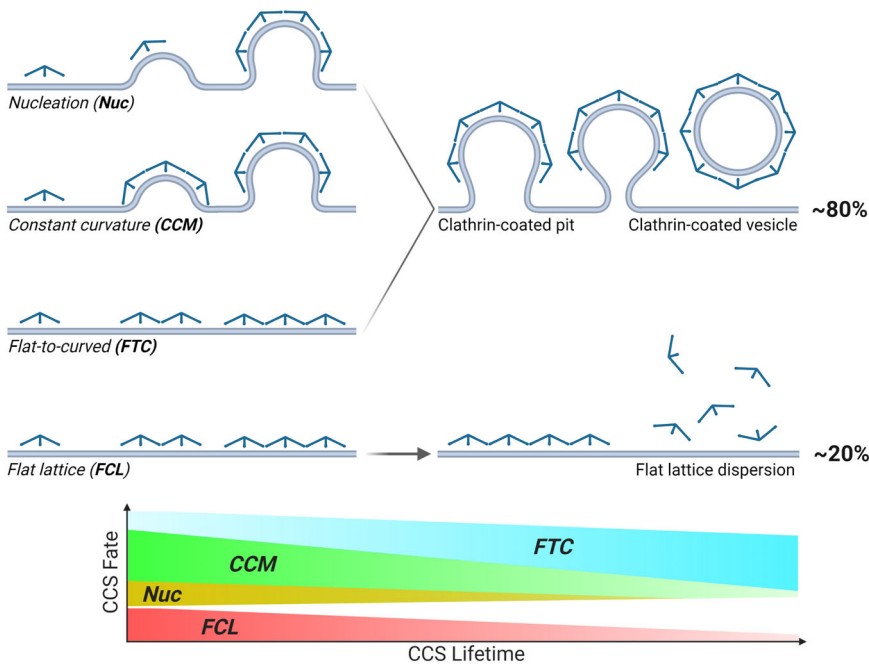

**Fig. 6 CME occurs through flexible model of CCVs formation, whereby the initiation of curvature and clathrin accumulation are not correlated and a single clathrin-centric model of vesicle formation cannot encompass the heterogeneity of CME dynamics.** Moreover, flexible model of endocytosis is consistent between different cell lines and cargos, and the balance of that model could be shifted by features such as PM tension, cargo size, or local concentration of adaptor proteins (Created with BioRender.com).

is corrected for chromatic aberration using the NanoGrid file and written by us image registration MATLAB script (IMGREG.m). Post-registration 647 nm channel images are divided by theirs unregistered version. Created ratio image is then used to measure the mean interpolation generated through image registration (measured as σ). First, ROI around the cell border is drawn, followed by measuring the mean signal within. In our data, on average σ = 1.01. This value is then used to apply a gaussian blur onto all the pictures of the 488 nm channel. Then Δz channel was generated by using written by us Fiji plugin (dz_channel_generator.ijm) that will automatically run further steps described in Supplementary Fig. 2b and save the data that can be then used for data analysis.

**EPI/STAR analysis.** STAR images were acquired as described above alternately with epifluorescence images using the Sola light source (Lumencor) and the 96362 ET GFP C180702 filter cube. An image set was acquired every 1.25 s with 200 ms exposure. Cohorts with a mean of 2 or less events per cell were discarded from quantifications in Supplementary Fig. 6c.

**Image processing.** Data were corrected and analyzed using Fiji (ImageJ, National Institutes of Health, Bethesda, MD), MATLAB version 2018b for CMEanalysis, and MATLAB 2020b.

**Monte Carlo simulation of CCV formation.** A model of STAR measurements of clathrin-coated vesicle formation was created using MATLAB 2020b. The clathrin-coated vesicle was represented as a sphere with a 50 nm radius and a surface area of 31415.9 nm². Clathrin locations were distributed uniformly across the sphere surface with a final density of 0.0637 CLCa/nm², or 200 CLCa per vesicle[61]. To model vesicle formation, a baseline was defined at z = 0 nm. The vesicle was placed entirely under the baseline and moved in increments of 1 nm along the z-axis until the entire vesicle was above the baseline. For each iteration, only clathrin locations at z > 0 contributed to the calculations. The center of mass (CM) for the forming vesicle was calculated at each z position, representing the ground truth. CLCa-STAR can occupy any proportion of the of the 200 CLCa locations. For a given vesicle and stage of formation, the theoretical Δz is determined by first calculating the sum of EGFP and iRFP713 intensities respectively generated at each CLCa-STAR location following Eqs. (1) and (2) for each location. Then, following Eqs. (3) and (4) and setting the initial intensity (I₀) to 100, the theoretical Δz is calculated. We simulated the stochastic recruitment of untagged (wild type) CLCa, and CLCa-STAR to a forming vesicle by randomly distributing CLCa-STAR on the vesicle in a defined percent of locations (100%, 70%, 50%, 30%, and 10%). We performed a Monte Carlo simulation of 100 different random distributions of molecules for each defined percent while calculating their Δz during vesicle formation.

**siRNA transfection and CLCa-STAR rescue**. 25 mm #1.5H glass coverslips (Thorlabs, CG15XH1) were washed in 200 proof (100%) ethanol (Decon Labs) and placed into 6-well polystyrene microplates (Falcon, 353046). Coverslips were washed three times with HBSS (Gibco, 14170161) and 3 ml of cell media was added and allow to equilibrate in a 37 °C and 5% CO$_2$ incubator. Cells were seeded at 70,000 cell/well and allowed to grow overnight. Prior to siRNA transfection cells were washed once with warmed full media. siRNA delivery was achieved using Lipofectamine RNAiMAX Transfection Reagent (Thermo Fisher Scientific, 13778030). Control Cos-2 cells were transfected with Negative Control siRNA (Qiagen, 1022076), and experimental cells were transfected with CLCa targeting siRNA (Santa Cruz Biotechnology, sc-35068) as follows (optimized protocol serves for one well of a 6-well dish)[43]: 5.5 μl of 20 μM siRNA and 10 μl of Lipofectamine RNAiMAX were added separately into 100 μl OptiMEM (Gibco, 11058021) and incubated for 5 min at RT. Then siRNA mixture was added to the Lipofectamine RNAiMAX mixture and gently mixed, followed by 10 min incubation at RT protected from light. The content of the tube was added dropwise evenly to the well. For western blot: 48 h post siRNA transfection, cells with silenced CLCa transfected with CLCa-STAR. Cell lysates were collected 24 h post CLCa-STAR transfection. For live-cell imaging: 48 h post siRNA transfection, control, and experimental cells were transfected with CLCa-STAR, and imaged the following day.

**Theory**. Equation (1) describes decay of TIRF evanescent wave, $I$ = Intensity, $I_0$ = intensity at 0 nm, $z$ = relative axial position, $d$ = wavelength dependent decay, described in Eq. (2), $\lambda$ = excitation wavelength, $n_1$ = refraction index of glass, $n_2$ = refraction index of the cell, and $\theta$ = angel of incidence.

$$I = I_0 e^{-\frac{z}{d}} \tag{1}$$

$$d = \frac{\lambda}{4\pi\sqrt{n_1^2\sin\theta^2 - n_2^2}} \tag{2}$$

Equation (3) describes the ratiometric approach behind STAR microscopy. $I^n_{488}$ = intensity at frame $n$ from TIRF 488 channel, $I^n_{647}$ = intensity at frame $n$ from TIRF 647 channel. Calculated intensity Ratio is the used in Eq. (4) to calculate the $\Delta z$ position of the molecule. $\gamma$ = constant described in Eq. (5).

$$R = \frac{I^n_{647}/I^n_{488}}{I^{AVG(1-10)}_{647}/I^{AVG(1-10)}_{488}} \tag{3}$$

$$\triangle z = \ln[R]\frac{1}{\gamma} \tag{4}$$

$$\gamma = \frac{d^{647} - d^{488}}{d^{647} \times d^{488}} \tag{5}$$

**Transferrin uptake assay**. 12 mm #1 glass coverslips (Electron Microscopy Science, 72231-01) were washed in 200 proof (100%) ethanol (Decon Labs) and placed into 6-well polystyrene microplates (Falcon, 353046). Three coverslips were placed per well for a total of 5 coverslips per transfection. Coverslips were washed three times with HBSS (Gibco, 14170161) and 3 ml of cell media was added and allowed to equilibrate in a 37 °C and 5% CO$_2$ incubator. Cells were seeded at 70,000 cell/well and left to grow overnight, or up to 70% confluency. Prior to transfection the cells were washed once with warm (37 °C) HBSS, followed with full fresh media replacement. Control cells were mock-transfected (no plasmid added), and experimental cells were transfected with CLCa-STAR. 24 h later, coverslips were transferred into a 24-well dish (one coverslip per well, Corning, 3524). Cells were washed with prewarmed HBSS, and Mock cells were incubated in serum-free FluoroBrite DMEM media (Gibco, A1896701), wherase CLCa-STAR over-expressing cells were incubated with serum-free FluoroBrite DMEM media supplemented with 5 μM biliverdin (co-factor for iRFP713; Cayman Chemical, 1925710), for at least 30 min. Media was then replaced with serum-free FluoroBrite DMEM media supplemented with 25 μg/ml[40] Transferrin-Alexa Fluor 568 conjugate[39] (TF-568, Thermo Fisher Scientific, T23365), and cells were incubated for 20, 15, 10, 5, and 0 min in a 37 °C. Then, 24-well dish was placed immediately on ice to stop internalization. Surface bound TF-568 was removed from the cells by a cold acid wash[51] (0.2 M Acetic Acid [Fisher Chemical, A38S-212], 0.2 M NaCl [Fisher Scientific, BP358], pH 2.5). Cells were then washed three times with cold DBPS (Thermo Fisher Scientific, 21-030-CV) and then fixed with 4% PFA (Electron Microscopy Sciences, 15710) for 20 min on ice. Cells were then washed three times with cold DBPS, mounted using VECTASHIELD mounting media (Vector Laboratories, H-1000) onto Microscope Slides (Fisherbrand, 12-550-15). Cells were imaged using Nikon Ti-2 microscope equipped with a 60× 1.49-NA objective, Sola light source (Lumencor), the 96362 ET GFP C180702 filter cube, the 96365 ET/mCh/TR C180338 filter cube, and ORCA-Flash 4.0 v3 scientific complementary metal-oxide-semiconductor camera chip (Hamamatsu). For analysis, pictures were background subtracted, cell bodies were outlined and mean fluorescent intensity of internalized TF-568 per cell was measured in Fiji.

**Transient transfection**. 25 mm #1.5H glass coverslips (Thorlabs, CG15XH1) were washed in 200 proof (100%) ethanol (Decon Labs) and placed into 6-well poly-styrene microplates (Falcon, 353046). Coverslips were washed three times with HBSS (Gibco, 14170161) and 3 ml of cell media was added and allowed to equi-librate in a 37 °C and 5% CO$_2$ incubator. Cells were seeded at 70,000 cell/well and left to grow overnight, or up to 70% confluency. Prior to transfection cells were washed once with warmed (37 °C) HBSS followed with full fresh media replace-ment. Cos-7 cells were transfected with the TransIT-LT1 (Mirus, MIR 2304) transfection reagent as follows (described protocol serves for one well of a 6-well dish): 100 μl of prewarmed (37 °C) RPMI 1640 (Gibco, 11875093) was placed into the 1.5 microcentrifuge tube (Fisherbrand, 05-408-129). Then 5 μl of room tem-perature (RT) TransIT-LT1 was gently added to the tube, gently mixed, and incubated for 5 min at RT. 1 μg of CLCa-STAR plasmid was added to the mixture and gently mixed. The mixture was incubated for 20 min at RT. Then, content of the tube was added dropwise evenly to the well. HUVECs were transfected with the TransIT-2020 (Mirus, MIR 5404) transfection reagent as follows (described pro-tocol serves for one well of a 6-well dish): 250 μl of prewarmed (37 °C) OptiMEM (Gibco, 11058021) was placed into the microcentrifuge tube. Then, 1 μg of CLCa-STAR plasmid was added to the OptiMEM and mixed, followed by addition of 1 μl of RT TransIT-2020. The mixture was then mixed and incubated for 20 min at RT. Then, the content of the tube was added dropwise evenly to the well. All cells were incubated with transfection mixture for 6 h at 37 °C and 5% CO$_2$ followed by full media replacement. Live-cell experiments were performed 24 h post-transfection. For STAR experiments three main criteria were used to select cells for imaging: (1) Cell footprint and spread area in TIRF, (2) Signal to noise in TIRF, and (3) Golgi visibility in Epi—since clathrin plays role in the vesicle formation at the trans-Golgi network (TGN) and it's localization to TGN is expected.

**Western-blot analysis and antibodies**. Proteins were extracted by cell lysis (RIPA buffer supplemented with cOmplete [Roche, 11697498001]) 48 h post-transfection, loaded at 50 μg per well (Laemmli Sample Buffer) on 10% polyacrylamide gel, and transferred to polyvinylidene difluoride (Bio-Rad, Immun-Blot LF PVDF) for 1 h at 100 V in 4 °C. Blots were blocked for 1 h at 4 °C with Intercept Blocking Buffer (Li-Cor, 927-66003) and immunoblotted with primary antibodies overnight at 4 °C, and secondary antibodies for 1 h at RT. Antibodies: rabbit anti-CLTA (Proteintech, 10852-1-AP; 1:1000), mouse anti-GAPDH (Cell Signaling Technology, 97166; 1:1000), goat anti-mouse IgG (Li-Cor, 925-68020, IRDye 680LT; 1:20,000) and goat anti-rabbit IgG (Li-Cor, 925-32211, IRDye 800CW; 1:15,000). All antibodies were diluted in Intercept Antibody Diluent (Li-Cor, 927-66003). Western blots were visualized using an Odyssey Image Station (Li-Cor) and the Odyssey Application Software (3.0, Li-Cor).

**Statistics and reproducibility**. All live-cell experiments were performed with three independent sets of transfections, unless stated otherwise. The number of total analyzed cells is indicated in figures legends. Western blots were performed with three independent sets of transfections. Each experiment was repeated inde-pendently with similar results. Figure legends contain the $n$ values for each data set, description of statistical test used, and $p$ values. All results are presented as mean ± SEM unless otherwise noted. Statistical calculations were performed in GraphPad Prism (Version 9.0.0).

**Reporting summary**. Further information on research design is available in the Nature Research Reporting Summary linked to this article.

## Data availability
Sample data sets generated in this study can be accessed by the GitHub database https://github.com/Mattheyses-Lab/Nawara_et_al._NatCommun_2022.git. The total raw and processed imaging dataset can be accessed by emailing the corresponding author. The data generated in this study are provided in the supplementary information. Source data are provided with this paper.

## Code availability
The modified CMEanalysis, MATLAB programs and Fiji scripts are available on the lab GitHub (https://github.com/Mattheyses-Lab/Nawara_et_al._NatCommun_2022.git).

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

## Acknowledgements

The authors support diversity and inclusiveness in science and believe strongly in equality of human rights. The authors would like to thank members of the Mattheyses Lab and Andrew Kowalczyk (Penn State) for their support and helpful discussion. We thank the UAB-Heflin sequencing core and UAB IT Research Computing. This work was

supported by NIH/NIGMS R01GM131099 to K.S. and A.L.M., NSF CAREER 1832100 to A.L.M. and American Heart Association 906086 to T.N.

## Author contributions

Conceptualization: T.N. and A.M.; methodology: T.N., Y.W., T.R., Y.H., K.S. and A.M.; validation: T.N. and A.M.; formal analysis: T.N.; investigation: T.N.; resources: Y.H., E.S., K.S. and A.M.; data curation: T.N.; writing—original draft: T.N. and A.M.; writing—reviewing and editing: T.N., T.R., E.S., K.S. and A.M.; visualization: T.N., Y.W.; supervision: T.R., K.S., E.S. and A.M.; funding acquisition: A.M., K.S. and T.N.

## Competing interests

The authors declare no competing interests.
