## [Peer Review File · Nature Communications]

Imaging vesicle formation dynamics supports the flexible model of clathrin-mediated endocytosisREVIEWER COMMENTS

Reviewer #1 (Remarks to the Author):

This manuscript describes the application of simultaneous two color TIRF imaging (STAR microscopy) to measure the axial localization of clathrin coats in living cells. This quantitative live cell approach allows the authors to address a question that has been very actively studied in the recent years: How is the assembly of the clathrin coat and the generation of membrane curvature linked? This question has critical implications for understanding the mechanism by which the clathrin coat generates membrane curvature during endocytosis. This question was first raised in the 80s, but due to technical challenges was not solved. Later the whole question was forgotten, and the textbooks adopted the “constant curvature” model without solid evidence. In the recent years, however, this question has been revived thanks to new imaging methods that are getting closer to spatial and temporal resolution required to answer the question. Still with the latest methods we have not been able to give a conclusive answer and thus the debate goes on. This manuscript provides solid evidence that, instead of one correct model, the process of clathrin-mediated endocytosis is more flexible, and can mediate curvature generation at least in two different ways (constant curvature and flat to curved models). This manuscript is methodologically solid and a significant contribution to the field. Below I list some suggestions I have for the authors to consider.

The authors interpret their Δz values as membrane curvature. However, this microscopy method does not measure curvature, but the protein localization along the z-axis. Now, I do agree with their interpretation, but it would be useful for the readers to have the reasoning behind it explained clearly. The authors emphasize the significant limits of electron microscopy data clearly in the manuscript, but then use the knowledge from EM about the relationship between the coat and membrane shapes to interpret their results without explicitly stating it.

Has the functionality of the double tagged clathrin that is used in the manuscript been tested in any way? The potential functionality should be at least discussed in the paper and ideally tested, if it has not been done yet.

It is possible that the tagged protein is not being accumulated with the same dynamics as the untagged protein. How would that affect the analysis? Is there a way to test this possibility?

As the tagged clathrin is expressed on top of endogenous untagged clathrin, there would be stochastic variation in the number of tagged clathrin molecules per endocytic site. How would this affect detection and the determining in which of the three classes an event belongs to? I am particularly thinking about the nucleation model. There would be a fraction of sites where the tagged molecules would be

randomly under represented. Also, the untagged molecule might get preferentially recruited. Would these events fall in the Nucleation model category? Could this be tested/controlled? (Obviously the best would be to use cells with full endogenous tagging of clathrin, but that, I would think, is beyond the scope of this manuscript.)

Fig1a: The curves in the panel 1a are not explained in the figure legend. These curves also don't very clearly explain the principle of the STAR microscopy. The intensity curves on the left seem identical, but the STAR curves are different and the figure makes it look like the different STAR curves come from identical raw data. The authors could try to illustrate the method a bit more clearly here.

Line 41: it is not just the axial resolution, but also the lateral resolution in light microscopy that is limiting its use for determining the morphology of clathrin structures.

Fig2a and later figures: The clathrin intensity cohort plots should have an actual time axes that shows the time, not just the cohort labels.

Line 143- : The idea of testing the nucleation model could be described more clearly here. As the curvature signal depends on accumulation of the labeled clathrin, it sounds paradoxical how the authors can detect curvature before clathrin accumulation?

This is explained on lines 157-160, but please, consider moving the explanation up where the reader is likely to wonder this point.

Lines 196-198: I disagree with this statement. I know numerous earlier papers from many labs that I could easily characterize as "comprehensive, high throughput and high-resolution quantification of clathrin coated vesicle formation in living mammalian cells". I don't think this sentence adds anything to the message of this manuscript.

Reviewer #2 (Remarks to the Author):

In this work the authors explore the structural transitions of clathrin-coated sites in cultured mammalian cells. By developing and using a multi-color differential TIRF excitation method (STAR) the author are

able to track the axial movements of the clathrin lattice with great precision in live cells. The resolution of these changes is impressive and allows for the careful analysis of curvature of the clathrin coat in the system. Specifically, the authors test two common and competing models of curvature. Supporting some past (and recent) work, the authors find that the pathways of clathrin curvature are heterogeneous and complex. Indeed, the curvature pathway falls somewhere in the middle of both common models. The authors conclude that the endocytic pathway for clathrin is very flexible, dynamic, and diverse. There are some surprising findings in this work. I think the information is useful to the field of membrane traffic and will help guide future experiments. The results do not present a “black or white” picture of endocytosis but I think this is an important concept about the behavior of cellular systems. The data are well analyzed and presented. I have some specific concerns that the authors should address to improve the manuscript. My specific comments follow.

1. The methods section could use some editing for clarity. As much of the paper relies on extensive and automated data processing it is important to have this section easy to understand and follow.

2. How is the STAR signal normalized? The bead experiment is nice but this is for an object that is resting directly on the coverslip. If the cell's (or portions of the cell) plasma membranes are at different axial heights, the signal ratios and their decays would be different depending on where they are in the TIRF field and the emitted photon's distance to the coverslip. Please expand on this analysis in the paper.

3. How did the authors decide on the delineation between flat and curved/spherical clathrin? At what point is something flat and curved? This would either bias the reported population measurements towards flat and curved depending on the cut-off point for curvature. Please expand on this justification and its relationship to past work.

4. The “nucleation” clathrin subclass is very difficult to understand. Here, dual-labeled clathrin is being used as a marker for curvature (which the authors point out) so I don't understand the justification for a class of curved sites that don't have clathrin before curvature. I would recommend either removing this class, combining it with another class, or further explaining the justification for this class. Indeed, it might be more helpful to image a lipid-anchored STAR probe and clathrin to further prove this curved nucleation class does not have clathrin. As it stands now, it is a bit confusing to the reader.

5. The justification to use EGF and VEGF stimulated cells is not clearly explained. Please explain why this was done as opposed to imaging non-starved unstimulated cells.

6. Are the long-lived clathrin sites that don't disappear analyzed by the automated tracking package? How many of these very long clathrin tracks are present in the data? If these are flat, then the lack of

these traces in the analysis could bias the reported population measurements. Please discuss this in the methods and text.

Reviewer #3 (Remarks to the Author):

In this paper, the authors present a promising methodological advance for measuring real-time axial displacement during clathrin-mediated endocytosis. The experiments are generally thorough and the data are mostly clear; however, the authors can clarify their model of biologically-relevant endocytic uptake. The shortcomings can largely be addressed through text changes, although a few simple experimental controls may be necessary to fully validate the results presented.

Fundamentally, scientists debating curvature generation during CME are split into two camps: the constant-curvature model (CCM) and the flat-to-curved transition model (FTC transition). The STAR method employed here promises to distinguish between the two competing models. While the authors explain the competing models well, the text then falls short of clearly distinguishing between the two models as events are visualized by STAR microscopy. This confusion is worsened by emphasizing putatively abortive CME sites, where clathrin builds up but curvature is never induced, confounding the takeaways from true endocytic sites where curvature is necessarily induced.

In the abstract, the authors state: “We show that clathrin accumulation drives curvature formation at shorter-lived clathrin-coated vesicles (CCVs), but clathrin undergoes a flat-to-curved transition at longer-lived CCVs.” This is a strong conclusion to draw, which seems based on one portion of Figure 3, where shorter-lived events are predominantly formed through CCM and longer-lived events favor FTC transition. These data do not support the conclusion that clathrin accumulation drives curvature formation; indeed, the two may be driven by the same mechanism. Also, this correlation is somewhat weak, with many events of each type appearing in each lifetime cohort. This result indicates likely functional relevance of the different mechanisms of curvature generation presented here, and that lifetime alone cannot explain the variation. The STAR method, applied to other experimental conditions, could be very useful in elucidating the specifics of these mechanistic differences like PM tension, cargo size, etc.

The text requires some clarifications and supplemental points addressed, as well:

1. While the calibration of the STAR method is sufficient to prove STAR's robustness, there is no discussion of the sensitivity of STAR or the expected error in the measurements. This leads the reader to wonder how confident the authors can really be about the extent of axial displacement generated. The axial displacement is often treated as a binary operator in the text (the vesicle is or is not raised); this could correspond to a displacement in z , indicating the actual vesicle height, but that is not given.
2. Is there a correction for expression level in the transient transfection of cells? Is there a correlation between expression level and proportion of event type/abortive events/FCLs? There is good evidence that expression level affects normal endocytic dynamics, which the authors acknowledge; however, the text does not indicate how expression level was controlled or chosen.
3. The threshold for valid detection of clathrin light chain and detection of curvature formation is 5 consecutive frames, or 2.5 seconds of detection (according to the 2 Hz imaging condition). This seems an arbitrary cutoff: does selection of a different threshold result in different conclusions drawn? Does the higher sensitivity of the clathrin detection over curvature detection confound the reported delay in curvature generation, or perhaps even explain it completely?
4. For iRFP713 to function, cells must be grown with added biliverdin. The authors do not test whether biliverdin affects endocytic dynamics, for instance by repeating the experiment without biliverdin and testing the lifetimes/overall cohort populations of the GFP detections of CLCa-STAR. This is unlikely to have any effect, but perhaps important to check, as iRFP713 is not a commonly-used fluorophore.
5. Some portions of figures add minimal value; for instance, panel 2E explains the frequency of curved vs flat CCSs per area per minute, but this is also essentially reported in panel 2D and is not functionally relevant to the model presented. The same type of panel is presented in figure 4.

A few experimental manipulations that alter the proportion of CCM vs FTC transition cohorts in a cell would greatly strengthen the conclusions drawn in this paper; however, that may be the topic of another manuscript.

REVIEWER COMMENTS

Reviewer #1 (Remarks to the Author):

This manuscript describes the application of simultaneous two color TIRF imaging (STAR microscopy) to measure the axial localization of clathrin coats in living cells. This quantitative live cell approach allows the authors to address a question that has been very actively studied in the recent years: How is the assembly of the clathrin coat and the generation of membrane curvature linked? This question has critical implications for understanding the mechanism by which the clathrin coat generates membrane curvature during endocytosis. This question was first raised in the 80s, but due to technical challenges was not solved. Later the whole question was forgotten, and the textbooks adopted the “constant curvature” model without solid evidence. In the recent years, however, this question has been revived thanks to new imaging methods that are getting closer to spatial and temporal resolution required to answer the question. Still with the latest methods we have not been able to give a conclusive answer and thus the debate goes on. This manuscript provides solid evidence that, instead of one correct model, the process of clathrin-mediated endocytosis is more flexible, and can mediate curvature generation at least in two different ways (constant curvature and flat to curved models). This manuscript is methodologically solid and a significant contribution to the field. Below I list some suggestions I have for the authors to consider.

Response: We want to thank the reviewer for their insightful analysis of the field and for recognizing the significant contribution of our manuscript. We especially thank the reviewer for suggestions which led to confirmation of the functionality of the CLCa-STAR probe with a series of new control experiments. Additionally, the impact of stochastic variation in recruitment of tagged and untagged molecules per endocytic site on detection and classification of observed coat bending dynamics has now been assessed theoretically and experimentally. Please see our detailed responses below.

R1 Comment 1: *The authors interpret their Δz values as membrane curvature. However, this microscopy method does not measure curvature, but the protein localization along the z-axis. Now, I do agree with their interpretation, but it would be useful for the readers to have the reasoning behind it explained clearly.*

R1 Response 1: Thank you for this comment. We agree and have clarified this important point when introducing STAR microscopy to the reader in the introduction.

“The axial resolution of STAR microscopy depends on object signal to noise ratio (S/N)¹, with up to 5nm axial resolution for high S/N objects. Fluorophores closer to the PM are brighter, and contribute to the higher axial resolution of STAR microscopy close to the PM. This makes STAR microscopy an ideal tool for studying the initiation of vesicle curvature formation. STAR microscopy reports the z-distribution of the dual-tagged protein. Given what is known about CME, when we image dual-tagged CLCa the STAR readout reflects PM curvature formation during vesicle assembly. Clathrin accumulation requires adaptor proteins, such as AP2, that link clathrin

and PM². As a vesicle invaginates there will be corresponding changes in the z-distribution of clathrin allowing CLCa-STAR to indirectly report PM shape and curvature formation.”

R1 Comment 2: *The authors emphasize the significant limits of electron microscopy data clearly in the manuscript, but then use the knowledge from EM about the relationship between the coat and membrane shapes to interpret their results without explicitly stating it.*

R1 Response 2: Thank you for identifying this oversight. We modified the discussion to directly address this point:

“STAR microscopy provides a dynamic context for the high-resolution images of clathrin coats with multiple architectures, compositions, and curvatures, reported by EM. Moreover, EM studies provide important context for our results, and lead us to the inclusive view of CCV formation while rejecting the dualism of opposing models. Our results put the clathrin structures detected by EM in context of living cells by showing the flexibility of clathrin-curvature interplay, and the involvement of FCLs in CME³⁻⁵.”

R1 Comment 3: *Has the functionality of the double tagged clathrin that is used in the manuscript been tested in any way? The potential functionality should be at least discussed in the paper and ideally tested, if it has not been done yet.*

R1 Response 3: To address the reviewer’s concern and test the functionality of the CLCa-STAR probe we performed a transferrin uptake assay. This is a standard assay used to assess disruptions to endocytosis by expression of tagged proteins⁶⁻⁸. This experiment showed that the overexpression of CLCa-STAR probe does not disrupt cargo internalization or act as a dominant negative. These data are now in **new Fig. 1f** and **new Supplementary Fig. 1**. We modified the results section to include these findings:

“We tested the functionality of CLCa-STAR in a transferrin uptake assay⁶⁻⁸ comparing wild type and CLCa-STAR expressing Cos-7 cells. No statistically significant differences were observed between matching time points, indicating expression of CLCa-STAR did not have a dominant effect on endocytic rates (Fig. 1f and Supplementary Fig. 1).”

We also would like to mention that two additional experiments were conducted which also speak to the functionality of the double tagged clathrin.

1. The lifetime distribution of clathrin-coated structures at the plasma membrane was compared between CLCa-STAR and the previously established probe CLCa-EGFP⁹. No differences were observed between the probes (*for details please see R3C7 and new Supplemental Fig. 5*).
2. We silenced endogenous CLCa followed by rescue with CLCa-STAR. When compared to control, there were no significant differences in in the lifetimes of flat or curved events or the curvature initiation dynamics (*for details see R1C4 and R1C5 and new Fig. 4b-i*)

R1 Comment 4: *It is possible that the tagged protein is not being accumulated with the same dynamics as the untagged protein. How would that affect the analysis? Is there a way to test this possibility?*

R1 Response 4: The reviewer makes an important point. In response, we conducted a knock-down rescue experiment¹⁰⁻¹² to address the influence of untagged clathrin light chain on detected CLCa-STAR probe accumulations, their ability to generate curvature, and their distribution through lifetime cohorts. In both control and CLCa silenced cells, we observed correct localization of CLCa-STAR probe at the plasma membrane, and curved and flat events forming throughout the experiment. The event distribution and the overall frequency were not significantly different.

These findings are presented in **new Fig. 4 b-e** and described in the result section:

“To experimentally test if recruitment dynamics of wild type and tagged CLCa influenced the lifetime distribution of events, their ability to undergo curvature, and detection of bending models reported by STAR microscopy, we used siRNA to knock down endogenous CLCa followed by expression of CLCa-STAR¹⁰⁻¹². We confirmed CLCa silencing and CLCa-STAR overexpression by western blot (Fig. 4b). We observed correct clathrin localization at the PM in both control siRNA and CLCa targeting siRNA transfected Cos-7 cells, and both curved and flat events were observed throughout the imaging time-lapse (Fig. 4c). Moreover, there were no significant differences in the distribution of events between matching lifetime cohorts (Fig. 4d) and the overall frequency of curved and flat events per cell (Fig. 4e).”

Further theoretical and experimental analysis of stochastic variation in tagged clathrin recruitment on reported by STAR microscopy findings is part of **R1C5**.

R1 Comment 5: *As the tagged clathrin is expressed on top of endogenous untagged clathrin, there would be stochastic variation in the number of tagged clathrin molecules per endocytic site. How would this affect detection and the determining in which of the three classes an event belongs*

to? I am particularly thinking about the nucleation model. There would be a fraction of sites where the tagged molecules would be randomly under represented. Also, the untagged molecule might get preferentially recruited. Would these events fall in the Nucleation model category? Could this be tested/controlled? (Obviously the best would be to use cells with full endogenous tagging of clathrin, but that, I would think, is beyond the scope of this manuscript.)

R1 Response 5: We agree that the presence of untagged clathrin could alter the classification of events based on bending dynamics, especially at the very early stages of vesicle lifetime. At this stage, clathrin intensity is low, and untagged clathrin could result in delayed curvature detection, once enough tagged clathrin are being recruited, or early curvature detection if enough tagged clathrin is recruited to detect curvature but not enough to count clathrin signal as positive. We addressed this experimentally and theoretically. We experimentally determined that all three initiation types, including nucleation, were detected at similar levels in cells with CLCa-STAR overexpression in the presence of WT CLCa or siRNA KD of WT CLCa (**new Fig. 4f-i** and replicated here). We expect the siRNA KD to remove any detection artifacts caused by untagged CLCa from the analysis since endogenous clathrin expression was minimal. The spectrum of bending dynamics was observed in both experimental conditions, with some minor non-statistically significant differences (**new supplementary Table 6** and **Reviewer Data 1**). This experimental result supports that the spectrum of bending dynamics we observed is not due to endogenous untagged CLCa and validates our experimental system. This is now described in the results section:

“Since endogenous clathrin expression was minimal, we expected to remove detection artifacts based on stochastic recruitment from the analysis of curvature initiation. We were particularly interested to investigate whether the presence of the untagged CLCa could explain events that fall into the nucleation category, as untagged CLCa could be driving curvature without a significant fluorescence signal. In both control and CLCa targeting siRNA treated cells we observed a spectrum of bending dynamics (Fig. 4f) and no correlation of curvature formation with CCS lifetime (Fig. 4g). Interestingly, events were distributed across all three bending models with or without the endogenous CLCa, with minor non-statistically significant differences (Fig. 4h, i, and Supplementary Tables 3-5). This data lets us conclude that the presence of endogenous CLCa in our system (Fig. 2 and 3) did not significantly alter CCSs formation dynamics, the ability to undergo curvature, or the classification of the events within the bending model.”

To further support these findings, we performed a Monte Carlo simulation of vesicles formation while varying the amount of CLCa-STAR tagged clathrin present in the system (**new Supplementary Fig. 8**, and **new Fig. 4a**). We show that CLCa-STAR probe measurements follow the ground truth with as less as 30% of tagged molecules contributing to vesicle formation. We modified the results section to reflect these findings.

“Our experimental system contained endogenous untagged CLCa, which could cause stochastic variation in the recruitment of tagged molecules to CCSs. This variation could impact the classification of events based on bending dynamics, especially at the early stages of vesicle formation. At this stage, clathrin intensity is low and a large percent of untagged clathrin at a CCS could result in delayed curvature detection. Alternately, if a majority of tagged clathrin is recruited, the intensity may be detected earlier, also confounding the quantification. To address how stochastic variation in the recruitment of CLCa-STAR and untagged CLCa alters Δz measurements, we performed a Monte Carlo simulation of STAR measurements during CCV formation. For each stage of vesicle formation, a ground truth center of mass and experimental Δz were calculated. The experimental Δz was calculated from the average intensity of randomly distributed CLCa-STAR molecules occupying 100%-10% of the available points on a Fibonacci sphere (Supplementary Fig. 8a). As expected, the SD of calculated Δz measurements was inversely correlated with the percentage of tagged molecules. Nevertheless, we found that STAR measurements followed the ground truth with as little as 30% of tagged molecules contributing to vesicle formation (Fig. 4a, and Supplementary Fig. 8b). The proportion of tagged:untagged CLCa in our experiments is estimated to be at least 50%. This suggests that the stochastic recruitment of tagged and untagged clathrin is not the driving factor underlying the variability reported here.”

R1 Comment 6: Fig1a: The curves in the panel 1a are not explained in the figure legend. These curves also don't very clearly explain the principle of the STAR microscopy. The intensity curves

on the left seem identical, but the STAR curves are different and the figure makes it look like the different STAR curves come from identical raw data. The authors could try to illustrate the method a bit more clearly here.

R1 Response 6: We appreciate the reviewer's concerns about illustrating STAR clearly, and we have modified **Figure 1 a, b** (replicated here) to address that. **New Fig 1a** illustrates models of vesicle formation and distinguishes between TIRF and STAR. The TIRF intensity signal has been adjusted to reflect anticipated differences. Yet these curves seem similar, as TIRF reports both the number of proteins and where they are within the evanescent field. This is now clarified with the “intensity indistinguishable” label. These axial curvature dynamics, though invisible by intensity measures, are revealed by STAR as indicated by the “curvature distinguishable” label. **New Fig. 1b** illustrates the physics behind STAR microscopy, showing the dependence of fluorescent intensity on axial position for two spectrally separated fluorophores. It also shows the design of the STAR probe and the dependence between CCS shape and measured ratio and Δz .

R1 Comment 7: Line 41: it is not just the axial resolution, but also the lateral resolution in light microscopy that is limiting its use for determining the morphology of clathrin structures.

R1 Response 7: Thank you, we have added this point to the main text.

“However, the limited lateral and axial resolution makes conventional fluorescence microscopy unsuitable for revealing the morphology of clathrin structures during the highly dynamic process of CME¹”

R1 Comment 8: Fig2a and later figures: The clathrin intensity cohort plots should have an actual time axes that shows the time, not just the cohort labels.

R1 Response 8: We thank the reviewer for this suggestion. The cohort plots were generated from CMEanalysis and built from interpolated tracks. Because of this, no linear time axis exists, and cohort labels are the standard way to display such data¹³⁻¹⁷. Therefore, we maintained cohort representation. This also has the benefit of making it easier to compare our data with other studies using CMEanalysis.

R1 Comment 9: Line 143- : The idea of testing the nucleation model could be described more clearly here. As the curvature signal depends on accumulation of the labeled clathrin, it sounds paradoxical how the authors can detect curvature before clathrin accumulation? This is explained on lines 157-160, but please, consider moving the explanation up where the reader is likely to wonder this point.

R1 Response 9: We apologize that the models were not clearly described. Following the reviewer suggestion, we have moved and modified the explanation to help the narrative. In the results, directly following the description of the 3 models we now include the following:

“...We note that since tagged clathrin serves as our readout, we can only observe nucleation sites where clathrin is present at a low level, possibly prior to its assembly into higher organized structures. The nucleation events reported here represent sites where the EGFP fluorescence intensity is still below the threshold to detect clathrin accumulation but the Δz is above the detection threshold sufficient to identify curvature formation.”

We have also modified our discussion to expand on our interpretation of the nucleation cohort and possible molecular mechanisms behind that model:

“BAR domain proteins, such as FCHO1/2^{18, 19}, can induce PM curvature through insertion of amphipathic helices or molecular crowding²⁰. Hence, curvature could develop prior to clathrin polymerization into higher organized structures at nucleation sites. Our results suggest the likely involvement of accessory proteins in early curvature generation^{12, 21}”

R1 Comment 10: Lines 196-198: I disagree with this statement. I know numerous earlier papers from many labs that I could easily characterize as “comprehensive, high throughput and high-resolution quantification of clathrin coated vesicle formation in living mammalian cells”. I don't think this sentence adds anything to the message of this manuscript.

R1 Response 10: Thank you for this critique, we have removed this statement from the text.

Reviewer #2 (Remarks to the Author):

In this work the authors explore the structural transitions of clathrin-coated sites in cultured mammalian cells. By developing and using a multi-color differential TIRF excitation method (STAR) the author are able to track the axial movements of the clathrin lattice with great precision in live cells. The resolution of these changes is impressive and allows for the careful analysis of curvature of the clathrin coat in the system. Specifically, the authors test two common and competing models of curvature. Supporting some past (and recent) work, the authors find that the pathways of clathrin curvature are heterogenous and complex. Indeed, the curvature pathway falls somewhere in the middle of both common models. The authors conclude that the endocytic pathway for clathrin is very flexible, dynamic, and diverse. There are some surprising findings in this work. I think the information is useful to the field of membrane traffic and will help guide future experiments. The results do not present a “black or white” picture of endocytosis but I think this is an important concept about the behavior of cellular systems. The data are well analyzed and presented. I have some specific concerns that the authors should address to improve the manuscript. My specific comments follow.

Response: We would like to thank the reviewer for recognizing the significance of our work and its contribution to the field of membrane traffic. We especially admire reviewer's appreciation of the non "black or white" conclusions presented in our work. We clarified our methods with additional figures and tables to reflect on our findings and provide a more in-depth explanation of the nucleation subclass more clearly. Please see below our detailed responses to the reviewer's concerns.

R2 Comment 1: *The methods section could use some editing for clarity. As much of the paper relies on extensive and automated data processing it is important to have this section easy to understand and follow.*

R2 Response 1: We apologize that automated data processing was not clearly described. We have addressed this in the revised manuscript with edits to the methods section, two new supplementary figures, and a new supplementary table.

New Supplementary Fig. 9 explains the automated analysis of the initiation of curvature formation. And **new Supplementary Fig. 10** explains the automated flat and curved events sorting. These figures each provide a detailed description of: **a** CMEanalysis outputs needed for automated data analysis, and post processing files description. **b** Data organization pre and post automated data processing using the given executable code. **c** Step-by-step explanation of data filtering and visualization.

New Supplementary Table 7 contains all the key parameters used for track detection and processing. Moreover, we provide executable code and example data. With the openly available code, sample data and **new Supplementary Table 7**, anyone should be able to reproduce our data processing. We have explained this in the under the "automated data processing" section of the methods:

"Automated Data processing. For automatic detection and tracking of endocytic events using CMEanalysis, TIRF 488, TIRF 647, and Δz channels for each cell were organized as previously published¹³ and submitted to the Cheaha supercomputer (UAB IT Research Computing) for processing. The modified CMEanalysis, other MATLAB programs, FIJI scripts, and sample data sets are available on the labs' GitHub and under the code availability section. The executable code and critical parameters are summarized in Supplementary Table 7. Here we analyzed CCSs, identified by CMEanalysis, that formed and left within the imaging time-lapse, which were 5s or longer, identified as a single track with a valid amount of missing signal (up to 13 frames = 3.9s; median gap length 1 frame), and positive for EGFP and iRFP713. Tracks identified by CMEanalysis as curvature positive were further processed using custom-written MATLAB programs (Supplementary Fig. 9a and 10a). Briefly, two CMEanalysis outputs, cell masks and detected and processed tracks, are automatically extracted and organized for further processing (Supplementary Fig. 9b and 10b). The distribution of Δz formation dynamics is automatically quantified and histograms and scatterplots correlating it to event lifetime are created by the MATLAB program `cme_wrapper.m`. Sorting of curved and flat events and track interpolation was achieved by the MATLAB program `cohort_wrapper.m`. The step-by-step details to this analysis are presented in Supplementary Fig. 9c and 10c and all the parameters used here are summarized in Supplementary Table 7."

Parameters used in automatic processing	
CMEanalysis for frame rate (FR) 0.3s	
Example code run for control group:	
<pre>>> data = loadConditionData('/data/user/tnawara/Data/Data analysis/AA_First paper/Revisions/siRNA/Exp3/CLCa_CTRL_exp3', {'Ch1', 'Ch2', 'Ch3'}, {'EGFP', 'iRFP713', 'dZ'}, 'Parameters', [1.49 60 6.45e-6]); >>[resCTRL, dataCTRL] = cmeAnalysis(data, 'ControlData', resCTRL, 'Overwrite', false, 'TrackingRadius', [1 3], 'TrackingGapLength', 13);</pre>	
Example code run for experimental group:	
<pre>>> data = loadConditionData('/data/user/tnawara/Data/Data analysis/AA_First paper/Revisions/siRNA/Exp3/CLCa_siRNA_exp3', {'Ch1', 'Ch2', 'Ch3'}, {'EGFP', 'iRFP713', 'dZ'}, 'Parameters', [1.49 60 6.45e-6]); >>[resKD, dataKD] = cmeAnalysis(data, 'ControlData', resCTRL, 'Overwrite', false, 'TrackingRadius', [1 3], 'TrackingGapLength', 13);</pre>	
1) 'Parameters'	[1.49 60 6.45e-6] / [NA Obj_Mag Camera_pix_size]
2) 'TrackingRadius'	[1 3]
3) 'TrackingGapLength'	13 (13*FR = 3.9s) filtered and valid tracks Median Gap length = 1
4) Start and end track buffer @ runTrackProcessing.m	[15 15] (15*0.3 = 4.5s)
5) Minimum track lifetime @ runTrackProcessing.m	6*0.3 = 1.8s
cme_wrapper.m @ dz_beginning.m	
1) Track length	≥ 5s
2) Single track with valid gaps	Category 1a (determined by CMEanalysis)
3) Is track iRFP713 and Δz positive	[1,1] (determined by CMEanalysis @ ProcessedTracks.mat -> tracks.significantSlave)
4) Numbers of consecutive positive frame over background to count signal beginning	5
5) Signal smoothing range for movmean	3
6) Frames below threshold to count signal as de novo	3
7) Determining whether signal higher than background	signal + background > 2*SD + background
8) Quality of iRFP713 signal	above threshold for more than 70% of EGFP signal
cohort_wrapper.m @ CCV_vs_FCL_graph_generator.m	
1) Track length	≥ 5s
2) Single track with valid gaps	Category 1a (determined by CMEanalysis)
3) Is track iRFP713 and Δz positive	[1,1] (determined by CMEanalysis @ ProcessedTracks.mat -> tracks.significantSlave)
4) numbers of consecutive positive frame over background to count signal beginning	5
5) Signal smoothing range for movmean	3
6) Frames below threshold to count signal as de novo	3
7) Last n frames have to be under threshold	3
8) Amount of Δz frames above threshold to count events as Δz positive	more than 30% of EGFP signal
9) Amount of Δz frames below threshold to count events as Δz negative	100%
10) Mean and SD of Δz negative signal	(-25nm ≤ signal ≤ 25 nm)

New Supplementary Table 7

New Supplementary Fig. 9

a**Extraction of data from CMEanalysis and description of output files**

CMEanalysis output needed from every analyzed cell for automatic data processing:
 1) cellmask.tif
 2) ProcessedTracks.mat

Extract automatically using:
 CME_extractor.m

Further data processing is designed to analyze data from one or more experimental repeats

Analyze automatically using:
 cohort_wrapper.m

Output files:
 Inside data folder:

- A) CellX_0.3s_dz_poz.mat
- Matrix containing Δz positive tracks per cell X
- B) CellX_0.3s_dz_neg.mat
- Matrix containing Δz negative tracks per cell X

Outside data folder

- C) data_Experiment_1dz_poz.mat
- Matrix containing Δz positive tracks per experiment
- D) data_Experiment_1dz_neg.mat
- Matrix containing Δz negative tracks per experiment
- C) data_total_dz_poz.mat
- Matrix containing Δz positive tracks per condition
- D) data_total_dz_neg.mat
- Matrix containing Δz negative tracks per condition

b**Data organization before and after automatic data processing**
cohort_wrapper.m

path = 'post_CMEanalysis_Test_Data_Cohort_Interp\Condition_1';

Output**c****Step-by-step filtering during automated data analysis**

Cell2_0.3s_Mask.tif
 Cell2_0.3s_Tracks.mat
 Cell7_0.3s_Mask.tif
 Cell7_0.3s_Tracks.mat
 Cell9_0.3s_Mask.tif
 Cell9_0.3s_Tracks.mat

n = 24761 tracks

Filtering:

- 1) Is the track lifetime 5s or longer? and
- 2) Is it a single tracks with valid gaps? and
- 3) Is the track iRFP713 positive?

n = 2907

Filtering:

- 1) Is the track within the cell mask?

n = 2872 filtered tracks

track processing

Binary matrices for each signal indicating whether above or below the threshold at each frame

Is signal higher than threshold?

- 1) Merging Start and end buffers with tracks
- 2) Smoothing: Signal, background and background deviation

Filtering:

- 1) Does Δz signal start below threshold?
- 2) Does clathrin signal start below threshold?
- 3) Does Δz signal end below threshold?
- 4) Does clathrin signal end below threshold?

5) Is Δz signal above the threshold for more than 30% of clathrin signal?

n = 268

Cohort_interpolation_STAR.m

Cohort interpolation:

- 1) Interpolate tracks into cohorts (ex. 10-20s, 20-40s, ...)
- 2) Plot cohorts that have more than 5 tracks

n = 56

5) Is Δz signal below threshold at all frames

6) Is Mean and SD of Δz signal 25nm or lower

7) Is Mean and SD of Δz signal -25nm or higher

New Supplementary Fig. 10

R2 Comment 2: How is the STAR signal normalized? The bead experiment is nice but this is for an object that is resting directly on the coverslip. If the cell's (or portions of the cell) plasma membranes are at different axial heights, the signal ratios and their decays would be different depending on where they are in the TIRF field and the emitted photon's distance to the coverslip. Please expand on this analysis in the paper.

R2 Response 2: The Δz reported is the STAR signal at time t_i normalized to the average of the first ten frames (t_1-t_{10}) of the imaging sequence to account for varying axial positions of the PM. Any tracks that overlap with the first ten frames are rejected from analysis. This normalization strategy has been tested previously, and there were no differences observed compared to normalizing to PM adjacent to the endocytic spot at the same time point ¹. Additionally something that may be of interest to the reviewer- multiple times we observed curved and flat clathrin puncta forming in close proximity, which eliminates the possibility of a global membrane fluctuation as the underlying driver of the axial height change. We have included some examples of those here (**Reviewer Data 2**, black filled arrows indicate clathrin accumulations that results in curvature formation, black empty arrows indicate adjacent clathrin accumulations that did not result in curvature formation). We have now clarified this important point in the result section as follows:

“To account for variations in the morphology of the PM, the reported Δz at time t_i was normalized to the average of the first ten frames (t_1-t_{10}) of the imaging sequence. This approach was previously shown to produce similar results to a time-matched normalization to adjacent PM¹.”

and

“Curved and flat events were detected adjacent to one another, indicating that STAR is reporting the dynamics of individual CCSs, and not large-scale PM fluctuations (Fig. 2b).”

R2 Comment 3: How did the authors decide on the delineation between flat and curved/spherical clathrin? At what point is something flat and curved? This would either bias the reported population measurements towards flat and curved depending on the cut-off point for curvature. Please expand on this justification and its relationship to past work.

R2 Response 3: We appreciate reviewer's comment. We are calculating the curvature formation pixel-by-pixel, and the resulting Δz image has both signal and noise. This approach gives us the power to use CMEanalysis as the unbiased classification system of the overall event curvature. Then events identified by CMEanalysis as curvature positive are further processed by our custom

written programs. We define curvature beginning as the moment when the curvature signal crosses the two standard deviations over the background for more than five consecutive frames, in agreement with common strategies in the field¹³. This delineation approach assures robust analysis, as it treats curvature formation event by event, while including signal noise, rather than a fixed threshold. We agree that the definition is important for the data, and have been careful not to draw conclusions based on the exact proportions of flat and curved events – only that both are represented. In response to **R3C6** we show that regardless of the cut-off point we observe the spectrum of bending dynamics, indicating the specific criteria is not artificially shifting the detected populations (**new Supplementary Fig. 7** and **new Supplementary Table 2.**; please see **R3C6** for details).

R2 Comment 4: *The “nucleation” clathrin subclass is very difficult to understand. Here, dual-labeled clathrin is being used as a marker for curvature (which the authors point out) so I don’t understand the justification for a class of curved sites that don’t have clathrin before curvature. I would recommend either removing this class, combining it with another class, or further explaining the justification for this class. Indeed, it might be more helpful to image a lipid-anchored STAR probe and clathrin to further prove this curved nucleation class does not have clathrin. As it stands now, it is a bit confusing to the reader.*

R2 Response 4: Thank you for this comment. We have carefully revised our language around the nucleation subclass throughout the manuscript. Moreover, in response to **R1C4** and **R1C5** we extended our analysis on the nucleation subclass with additional experiments and modeling. Briefly, we silenced wild-type CLCa using siRNA and then expressed CLCa-STAR allowing us to test for potential detection artifacts in cells expressing both endogenous and STAR tagged CLCa. Upon knockdown of WT CLCa, we still observed a spectrum of bending dynamics, which includes the nucleation sites (**new Fig. 4b-i**). We have also reanalyzed data by making the signal detection cut-off point more or less stringent and we still observed nucleation sites (for details please see **R3C6**, **new Supplementary Fig. 8**). Lastly, we have performed mathematical modeling to test how stochastic variation of tagged and untagged molecules would influence curvature detection (**R1C5**, **new Fig. 4a** and **new Supplementary Fig. 8**). We believe that the presence of endocytic accessory proteins at the nucleation sites could explain this subclass and we have modified our discussion to reflect on this finding and provide a more in-depth mechanistic explanation. The updated introduction of the nucleation subclass in the results reads as follows:

“We note that since tagged clathrin serves as our readout, we can only observe nucleation sites where clathrin is present at a low level, possibly prior to its assembly into higher organized structures. The nucleation events reported here represent sites where the EGFP fluorescence intensity is still below the threshold to detect clathrin accumulation but the Δz is above the detection threshold sufficient to identify curvature formation.”

R2 Comment 5: *The justification to use EGF and VEGF stimulated cells is not clearly explained. Please explain why this was done as opposed to imaging non-starved unstimulated cells.*

R2 Response 5: We apologize that this was not clearly explained in the text. Both non-starved unstimulated cells and starved cells stimulated with only one ligand are common strategies used

in the field ²²⁻²⁴. Our starvation/stimulation strategy assures synchronization of the system and improves TIRF imaging of endocytosis. Following imaging of Cos-7 after EGF stimulation, we choose to image HUVECs after VEGF stimulation for consistency. The text has been updated and now reads as follows:

“Next, to synchronize ligand stimulated endocytosis and optimize TIRF imaging^{23, 24}, transfected cells were serum starved for at least 30 min followed by stimulation with epidermal growth factor (EGF, 100 ng/ml)²² immediately prior to imaging.”

And

“We used VEGF as the ligand to allow comparisons with our findings in Cos-7 cells, as VEGF is one of the key growth factors in this cell line²⁵”

R2 Comment 6: *Are the long-lived clathrin sites that don't disappear analyzed by the automated tracking package? How many of these very long clathrin tracks are present in the data? If these are flat, then the lack of these traces in the analysis could bias the reported population measurements. Please discuss this in the methods and text.*

R2 Response 6: Thank you for this comment. Long-lived clathrin structures are not the focus of this study and

were not analyzed. Events that do not disappear during the imaging sequence have an undefinable lifetime and their fate can only be assumed. Hence, the classification of such structures is speculative by definition and prone to error. However, we do observe interesting dynamics that are not part of our analysis, for example clathrin accumulations above the diffraction limit. Those types of structures would require a more in-depth and lower throughput analysis, which could be a focus of future publications. An example of such accumulation, replicated here shows that a part of a bigger clathrin lattice bends to create a curved structure (**Reviewer Data 3**, white arrows point towards the mentioned event). This type of structure has been reported by others²⁶. We have modified our results and discussion to clarify that we only analyze diffraction limited puncta that appear and disappear within the timeframe of the experiment, and longer lived and larger structures are outside the scope of this study.

In the results:

“To quantify the dynamics of different clathrin structures and their relative contributions to CME, we performed a high throughput screening of 1948 *de novo* CLCa-STAR accumulations which appeared and disappeared within the imaging sequence from 13 cells using CMEanalysis¹³”

And in the discussion

“It is worth noting that we focused on diffraction limited, *de novo*, and complete CME events, with a total duration shorter than 100s. While we observed interesting curvature formation dynamics in larger structures, their high throughput analysis is difficult. Lower throughput analysis of such structures may reveal even more complex pathways of vesicle formation, such as those identified by atomic force microscopy²⁶.”

Reviewer #3 (Remarks to the Author):

In this paper, the authors present a promising methodological advance for measuring real-time axial displacement during clathrin-mediated endocytosis. The experiments are generally thorough and the data are mostly clear; however, the authors can clarify their model of biologically-relevant endocytic uptake. The shortcomings can largely be addressed through text changes, although a few simple experimental controls may be necessary to fully validate the results presented. Fundamentally, scientists debating curvature generation during CME are split into two camps: the constant-curvature model (CCM) and the flat-to-curved transition model (FTC transition). The STAR method employed here promises to distinguish between the two competing models. While the authors explain the competing models well, the text then falls short of clearly distinguishing between the two models as events are visualized by STAR microscopy. This confusion is worsened by emphasizing putatively abortive CME sites, where clathrin builds up but curvature is never induced, confounding the takeaways from true endocytic sites where curvature is necessarily induced.

Response: We thank the reviewer for their critique. We have performed new experiments to show that biliverdin does not disrupt endocytic dynamics. We also reanalyzed our data with a range of signal detection criteria to show that choice of signal detection does not explain the variation of bending dynamics. Moreover, we have carefully revised our language and clarified the highlighted shortcomings. We also hoped for a clear distinction between competing models of vesicle formation, but the data leads us to support the “flexible” model, which has been recently posited and is gaining in popularity. We leave open the possibility that underlying variation could be explained by biophysical factors, highlighted by the reviewer, such as plasma membrane tension or cargo size, etc. This avenue, as recognized by the reviewer, will be the scope of future studies. Please see our detailed responses to comments below.

R3 Comment 1: *In the abstract, the authors state: “We show that clathrin accumulation drives curvature formation at shorter-lived clathrin-coated vesicles (CCVs), but clathrin undergoes a flat-to-curved transition at longer-lived CCVs.” This is a strong conclusion to draw, which seems based on one portion of Figure 3, where shorter-lived events are predominantly formed through CCM and longer-lived events favor FTC transition. These data do not support the conclusion that clathrin accumulation drives curvature formation; indeed, the two may be driven by the same mechanism.*

R3 Response 1: Thank you for pointing that out. We have clarified our language in the abstract and throughout the text to better reflect our findings. In the abstract we now state:

“We show clathrin accumulation is preferentially simultaneous with curvature formation at shorter-lived clathrin-coated vesicles (CCVs) but favors a flat-to-curved transition at longer-lived CCVs.”

R3 Comment 2: *Also, this correlation is somewhat weak, with many events of each type appearing in each lifetime cohort. This result indicates likely functional relevance of the different mechanisms of curvature generation presented here, and that lifetime alone cannot explain the variation. The STAR method, applied to other experimental conditions, could be very useful in elucidating the specifics of these mechanistic differences like PM tension, cargo size, etc.*

R3 Response 2: Thank you for this comment. We have now clarified our language around the interpretation of our results. We stress that the lifetime of the event does not explain the variation of coat curvature initiation in an obvious linear fashion. However, we observed some interesting features after lifetime grouping and model classification. The main conclusion is that events shorter than 20s prefer the CCM, while events longer than 20s are more likely to undergo a FTC transition. We agree with the reviewer that lifetime alone can not explain the variation. The results have been modified as follows:

“Since we measured a range of behaviors in when clathrin and Δz crossed the threshold, we asked if there was a correlation with the total clathrin lifetime. We hypothesized that the longer persistence of clathrin at PM could indicate delayed initiation of curvature formation⁵. Surprisingly, we found no obvious correlation (Fig. 3d). To further analyze this data, we grouped endocytic events based on clathrin lifetime and proposed models of vesicle formation. While events were present in all combinations, this revealed some interesting features. We found that short-lived endocytic events (<20s) were formed predominantly through the CCM, while longer events (>20s) favored the FTC transition (Fig. 3e, f, and Supplementary Table 1). The proportion of nucleation was smaller and favored short-lived events.”

Testing whether PM tension or cargo size could explain the variation would be an excellent future application of STAR and we have included this in the discussion:

“The ability of STAR to measure nanometer axial dynamics will help define the roles of distinct flat and curved clathrin structures. What underlines the variation in PM bending dynamics and whether we can shift the preferred route of the model remains to be investigated. Future research using STAR microscopy will define how the recruitment of accessory proteins and the role of biophysical parameters contribute to vesicle formation and how different clathrin dynamics impact signaling and cellular homeostasis.”

R3 Comment 3: *The text requires some clarifications and supplemental points addressed, as well: 1. While the calibration of the STAR method is sufficient to prove STAR’s robustness, there is no discussion of the sensitivity of STAR or the expected error in the measurements. This leads the reader to wonder how confident the authors can really be about the extent of axial displacement generated.*

R3 Response 3: Thank you for this suggestion. We have expanded on STAR microscopy sensitivity and expected error in measurements when introducing the technique to the reader. It was previously found¹ that the sensitivity of STAR microscopy depends on the signal to noise ratio (S/N), and for the bright objects axial resolution can be as low as 5 nm. Because of the exponential decay of the evanescent field, fluorophores closer to the coverslip will be brighter, providing better axial sensitivity. We have accessed the S/N of curvature positive events classified in each bending model and show lack of significant differences between them. That assures that each event was classified with similar axial accuracy (**Reviewer Data 4**). We include the following statement in the introduction:

“The axial resolution of STAR microscopy depends on object signal to noise ratio (S/N)¹, with up to 5nm axial resolution for high S/N objects. Fluorophores closer to the PM are brighter, and contribute to the higher axial resolution of STAR microscopy close to the PM. This sensitivity makes STAR microscopy an ideal tool for studying the initiation of vesicle curvature formation.”

Moreover, we performed a modeling analysis of STAR measurements in correlation to the amount of fluorescent signal and we found that STAR measurements follow the ground truth with as little as 30% of tagged molecules contributing to vesicle formation (**new Fig 4a** and **new Supplementary Fig. 8** (for details please see response to **R1C5**).

R3 Comment 4: *The axial displacement is often treated as a binary operator in the text (the vesicle is or is not raised); this could correspond to a displacement in z, indicating the actual vesicle height, but that is not given.*

R3 Response 4: Thank you for this comment. As published before¹, the axial resolution of STAR increases with progressive object translocation away from the coverslip, with 8nm resolution close to plasma membrane and 43nm for objects 300 nm away from the coverslip. This assures sensitive detection of initial curvature formation but might result in skewed interpretation of actual vesicle height due to the convoluted 3D structure of forming vesicle. With further data analysis, supported by extensive mathematical modeling, we are investigating how able to correct for the convolution of CLCa distribution, vesicle morphology, and STAR sensitivity. We hope these efforts will allow us to provide absolute vesicle sizes as well as the rates of curvature formation in the future, but at this point in time such quantification would be premature. In this manuscript, where we focus on discriminating endocytosis from flat clathrin and the initiation of curvature in vesicle formation, we feel our conclusions do not require a quantification of actual vesicle height.

R3 Comment 5: *Is there a correction for expression level in the transient transfection of cells? Is there a correlation between expression level and proportion of event type/abortive events/FCLs? There is good evidence that expression level affects normal endocytic dynamics, which the authors acknowledge; however, the text does not indicate how expression level was controlled or chosen.*

R3 Response 5: Thank you for this comment. In general, we imaged cells with similar intensities, avoiding those that were extremely bright. This is of course subjective and non-quantitative. For STAR experiments three main criteria were used to select cells for imaging: 1) Cell footprint and spread area in TIRF, 2) Signal to noise in TIRF, and 3) Golgi visibility in Epi - since clathrin plays role in the vesicle formation at the trans-Golgi network (TGN) it's localization to TGN is expected. We have modified the methods section to state these selection criteria:

“For STAR experiments three main criteria were used to select cells for imaging: 1) Cell footprint and spread area in TIRF, 2) Signal to noise in TIRF, and 3) Golgi localization of CLCa-STAR in Epi (expected because clathrin plays role in the vesicle formation at the trans-Golgi network).”

To directly address the concern, we conducted a correlation analysis to between EGFP intensity and proportion of curved/flat events in the cells we imaged. There was no significant correlation, indicating expression level did not strongly impact the proportion of flat/curved events (**new Supplementary Fig. 4**). In the results we added:

“The ratio of flat-to-curved CCSs in individual cells was not correlated to EGFP intensity at t_0 , indicating the relative proportions of reported behaviors was not due to CLCa-STAR expression level (Supplementary Fig. 4).”

We also conducted STAR imaging following siRNA KD of endogenous CLCa which reproduced results non statistically different from control in terms of flat/curved ratio and curvature initiation (see **new Fig. 4b-i**, and our response to **R1C4** and **R1C5**).

R3 Comment 6: *The threshold for valid detection of clathrin light chain and detection of curvature formation is 5 consecutive frames, or 2.5 seconds of detection (according to the 2 Hz imaging condition). This seems an arbitrary cutoff: does selection of a different threshold result in different conclusions drawn? Does the higher sensitivity of the clathrin detection over curvature detection confound the reported delay in curvature generation, or perhaps even explain it completely?*

R3 Response 6: The reviewer makes an interesting point. We tested this possibility by both increasing and decreasing the sensitivity of the detection cut-off. We found that adjusting the sensitivity could not explain the spectrum of bending dynamics, as events in each subclass remained in all tested detection cutoffs. We also noted that as the cutoff in increased, the number of puncta that fulfill the criteria decreased, as anticipated. These data are now included as **new Supplementary Fig. 7** and **new Supplementary Table 2**. We have included these findings in the results:

“It is possible the number of consecutive frames above threshold defined for CLCa and curvature detection could alter our findings and possibly explain the reported spectrum of bending dynamics. We tested this by adjusting the required number of sequential frames above the threshold required to define the beginning of intensity and curvature from 2-20 frames (0.6-6 s; Supplementary Fig. 7; Supplementary Table 2). As the signal detection threshold increased, the total number of puncta that fulfill the criteria decreased, as anticipated. However, the spectrum of bending dynamics reported at 5 frames (1.5 s) was present throughout and not explained by the detection sensitivity.”

R3 Comment 7: 4. For iRFP713 to function, cells must be grown with added biliverdin. The authors do not test whether biliverdin affects endocytic dynamics, for instance by repeating the experiment without biliverdin and testing the lifetimes/overall cohort populations of the GFP detections of CLCa-STAR. This is unlikely to have any effect, but perhaps important to check, as iRFP713 is not a commonly-used fluorophore.

R3 Response 7: We thank the reviewer for this suggestion. Following reviewer’s recommendation, we have tested the distribution of clathrin coated structures with or without the addition of biliverdin to the starvation media for 30 minutes before imaging. We found no significant differences in the lifetime of clathrin-coated structures or the overall frequency of events. We also compared the endocytic dynamics of CLCa-STAR with and without biliverdin to the previously established CLCa-EGFP probe⁹. We found no statistical differences between matching lifetime cohorts or the overall frequency of events when compared to CLCa-STAR expressing cells. We note that the distribution of events across lifetime cohorts is not Raleigh distributed. This was expected as our analysis was optimized to support 3 color detection and in our experiential system the same track needs to be positive for EGFP, iRFP713 and Δz (total of three different

New Supplementary Fig. 7

Mean % tracks	[5-20s]			(20-50s)			>50s			Median for all events	Mode for all events
	Nuc	CCM	FTC	Nuc	CCM	FTC	Nuc	CCM	FTC		
2 (0.6s)	10.21 (+2.66)	31.98 (-0.8)	15.08 (+0.19)	3.15 (+1.30)	9.91 (+3.30)	15.68 (-4.76)	1.71 (+0.74)	2.93 (+0.38)	9.35 (-3.01)	2.7s (-0.9s)	[0.5-1.5s] (-1s)
5 (1.5s)	7.55	32.78	14.89	1.85	6.60	20.44	0.97	2.56	12.37	3.6s	[1.5-2.5s]
7 (2.1s)	7.21 (-0.34)	32.12 (-0.66)	13.04 (-1.84)	1.78 (-0.07)	6.31 (-0.29)	22.14 (-1.71)	1.04 (+0.07)	2.44 (-0.12)	13.91 (+1.54)	3.9s (+0.3s)	[1.5-2.5s] (ND)
11 (3.3s)	6.85 (-0.70)	30.95 (-1.82)	10.59 (-4.30)	1.54 (-0.31)	6.21 (-0.40)	23.85 (-3.41)	0.94 (-0.03)	2.76 (+0.21)	16.32 (+3.95)	3.8s (+0.3s)	[1.5-2.5s] (ND)
20 (6s)	6.88 (-0.67)	22.16 (-10.62)	9.53 (-5.36)	2.64 (+0.79)	6.81 (+0.20)	25.38 (-4.94)	1.74 (+0.77)	2.22 (-0.34)	22.65 (+10.28)	5.7s (+2.1s)	[-0.5-0.5s] (-2s)

New Supplementary Table 2

signals) to be considered a clathrin-coated vesicle. The data in this experiment contained only signal from EGFP in all conditions (without biliverdin there is negligible iRFP emission). These data are now presented in **new Supplementary Fig 5**, and the following was added to the result section:

“To ensure our findings with the CLCa-STAR probe reflect endocytic dynamics, we conducted control experiments to test the addition of the iRFP713 fluorophore to the established CLCa-EGFP⁹ probe and the impact of its cofactor biliverdin. We compared the localization and dynamics of CLCa-STAR in media supplemented with or without biliverdin to CLCa-EGFP (Supplementary Fig. 5a). Using CMEanalysis, CCSs were identified based on the EGFP signal only. We found no significant difference in the lifetime distribution of endocytic events or the overall event frequency (Supplementary Fig. 5b, c), indicating that dual tagging CLCa and the addition of biliverdin do not disrupt the overall dynamics of clathrin structures at the PM.”

R3 Comment 8: 5. Some portions of figures add minimal value; for instance, panel 2E explains the frequency of curved vs flat CCSs per area per minute, but this is also essentially reported in panel 2D and is not functionally relevant to the model presented. The same type of panel is presented in figure 4.

R3 Response 8: We appreciate reviewers comment. Although based on the same data, we feel these graphs report two different important points: 1) Fig. 2d looks at all individual events, designated as curved or flat, and reports the event lifetime; 2) Fig. 2e presents the variation in flat and curved events between cells, and is important for data reproducibility. Hence, we believe that reporting both graphs is informative for the field and important to support our claims.

R3 Comment 9: A few experimental manipulations that alter the proportion of CCM vs FTC transition cohorts in a cell would greatly strengthen the conclusions drawn in this paper; however, that may be the topic of another manuscript.

R3 Response 9: We appreciate the reviewer’s interest in deepening our understanding of the reasons underlying the variation of clathrin-coat bending dynamics. We are now in the process of testing biophysical parameters, such as osmotic pressure and concertation of endocytic accessory proteins, on puncta shape. However, this work remains preliminary and we believe is outside the scope of this study. We plan to address these exciting questions with further publications, as suggested and highlighted in the discussion as included in **R3C2**.

References:

1. Stabley, D.R., Oh, T., Simon, S.M., Mattheyses, A.L. & Salaita, K. Real-time fluorescence imaging with 20 nm axial resolution. *Nat Commun* **6**, 8307 (2015).
2. Kirchhausen, T. Three ways to make a vesicle. *Nature Reviews Molecular Cell Biology* **1**, 187-198 (2000).
3. Sochacki, K.A., Dickey, A.M., Strub, M.P. & Taraska, J.W. Endocytic proteins are partitioned at the edge of the clathrin lattice in mammalian cells. *Nat Cell Biol* **19**, 352-361 (2017).
4. Sochacki, K.A. *et al.* The structure and spontaneous curvature of clathrin lattices at the plasma membrane. *Dev Cell* (2021).
5. Sochacki, K.A. & Taraska, J.W. From Flat to Curved Clathrin: Controlling a Plastic Ratchet. *Trends Cell Biol* **29**, 241-256 (2019).
6. Doyon, J.B. *et al.* Rapid and efficient clathrin-mediated endocytosis revealed in genome-edited mammalian cells. *Nat Cell Biol* **13**, 331-337 (2011).
7. Lyszkiewicz, M. *et al.* Human FCHO1 deficiency reveals role for clathrin-mediated endocytosis in development and function of T cells. *Nat Commun* **11**, 1031 (2020).
8. Mino, R.E., Chen, Z., Mettlen, M. & Schmid, S.L. An internally eGFP-tagged alpha-adaptin is a fully functional and improved fiduciary marker for clathrin-coated pit dynamics. *Traffic* **21**, 603-616 (2020).
9. Rappoport, J.Z., Simon, S.M. & Benmerah, A. Understanding living clathrin-coated pits. *Traffic* **5**, 327-337 (2004).
10. Tsygankova, O.M. & Keen, J.H. A unique role for clathrin light chain A in cell spreading and migration. *J Cell Sci* **132** (2019).
11. Chen, P.H. *et al.* Crosstalk between CLCb/Dyn1-Mediated Adaptive Clathrin-Mediated Endocytosis and Epidermal Growth Factor Receptor Signaling Increases Metastasis. *Dev Cell* **40**, 278-288 e275 (2017).
12. Mettlen, M. *et al.* Endocytic accessory proteins are functionally distinguished by their differential effects on the maturation of clathrin-coated pits. *Mol Biol Cell* **20**, 3251-3260 (2009).
13. Aguet, F., Antonescu, C.N., Mettlen, M., Schmid, S.L. & Danuser, G. Advances in analysis of low signal-to-noise images link dynamin and AP2 to the functions of an endocytic checkpoint. *Dev Cell* **26**, 279-291 (2013).
14. He, K. *et al.* Dynamics of Auxilin 1 and GAK in clathrin-mediated traffic. *J Cell Biol* **219** (2020).
15. Pascolutti, R. *et al.* Molecularly Distinct Clathrin-Coated Pits Differentially Impact EGFR Fate and Signaling. *Cell Rep* **27**, 3049-3061 e3046 (2019).
16. Srinivasan, S. *et al.* A noncanonical role for dynamin-1 in regulating early stages of clathrin-mediated endocytosis in non-neuronal cells. *PLoS Biol* **16**, e2005377 (2018).
17. Kadlecova, Z. *et al.* Regulation of clathrin-mediated endocytosis by hierarchical allosteric activation of AP2. *J Cell Biol* **216**, 167-179 (2017).
18. Day, K.J. *et al.* Liquid-like protein interactions catalyse assembly of endocytic vesicles. *Nat Cell Biol* **23**, 366-376 (2021).
19. Henne, W.M. *et al.* FCHO proteins are nucleators of clathrin-mediated endocytosis. *Science* **328**, 1281-1284 (2010).
20. Simunovic, M., Evergren, E., Callan-Jones, A. & Bassereau, P. Curving Cells Inside and Out: Roles of BAR Domain Proteins in Membrane Shaping and Its Cellular Implications. *Annual Review of Cell and Developmental Biology* **35**, 111-129 (2019).
21. McMahon, H.T. & Boucrot, E. Molecular mechanism and physiological functions of clathrin-mediated endocytosis. *Nat Rev Mol Cell Biol* **12**, 517-533 (2011).

22. Mattheyses, A.L., Atkinson, C.E. & Simon, S.M. Imaging single endocytic events reveals diversity in clathrin, dynamin and vesicle dynamics. *Traffic* **12**, 1394-1406 (2011).
23. Rappoport, J.Z. & Simon, S.M. Endocytic trafficking of activated EGFR is AP-2 dependent and occurs through preformed clathrin spots. *J Cell Sci* **122**, 1301-1305 (2009).
24. Wang, X. *et al.* DASC, a sensitive classifier for measuring discrete early stages in clathrin-mediated endocytosis. *Elife* **9** (2020).
25. Gaengel, K. & Betsholtz, C. Endocytosis regulates VEGF signalling during angiogenesis. *Nat Cell Biol* **15**, 233-235 (2013).
26. Tagiltsev, G., Haselwandter, C.A. & Scheuring, S. Nanodissected elastically loaded clathrin lattices relax to increased curvature. *Sci Adv* **7** (2021).

REVIEWERS' COMMENTS

Reviewer #1 (Remarks to the Author):

The authors have successfully answered all my concerns by clarifying the text and by performing thorough control experiments, particularly to test the functionality of the CLCa-STAR probe. I recommend accepting the manuscript.

Reviewer #2 (Remarks to the Author):

The authors have addressed all my previous comments. It is an excellent and interesting paper. I have one major and several very minor suggestions to clarify several aspects of the paper. Please address these in the text. This is a wonderful, well analyzed, and important paper.

My major comment is:

Line 265. Do these cells not have CLCb? How would this change the interpretation of these results? Please discuss and clarify this point in the text and references. I don't expect additional experiments or analysis, only a clarification or discussion of this caveat in the text.

My minor suggestions are:

Line 57: Flat lattices have been shown to contain pentagons (Sochacki et al. 2021). Possibly edit to "proposed by some models to contain mostly" or something similar.

Line 154: Please indicate here that pre-existing clathrin structures and longer-lived structure were excluded from the cohort analysis. The authors do discuss this later in the paper but I think this would focus the narrative of the work.

Line 199: Consider edit to "substantial clathrin accumulation" or similar for clarity.

Linw 200: Consider edit to “robust accumulation” or similar for clarity.

Line 206: Consider edit to “unambiguously classify the site as a clathrin accumulation site” or similar for clarity.

Line 207: Consider edit of “identify” to “measure”.

Reviewer #3 (Remarks to the Author):

I'm overall satisfied with the responses to my reviews. I still think still that the authors could better flesh out what they mean by the flexible model of CME during their introduction.

REVIEWERS' COMMENTS

Reviewer #1 (Remarks to the Author):

The authors have successfully answered all my concerns by clarifying the text and by performing thorough control experiments, particularly to test the functionality of the CLCa-STAR probe. I recommend accepting the manuscript.

Thank you.

Reviewer #2 (Remarks to the Author):

The authors have addressed all my previous comments. It is an excellent and interesting paper. I have one major and several very minor suggestions to clarify several aspects of the paper. Please address these in the text. This is a wonderful, well analyzed, and important paper.

Thank you.

My major comment is:

Line 265. Do these cells not have CLCb? How would this change the interpretation of these results? Please discuss and clarify this point in the text and references. I don't expect additional experiments or analysis, only a clarification or discussion of this caveat in the text.

Done @ line 261-263 and 276,

“Our experimental system contained endogenous untagged CLCa and its isoform CLCb^{48, 49}. The CLC isoforms randomly associate with clathrin triskelia^{48, 50} and stochastic variation in the recruitment of tagged molecules to CCSs could impact the classification of events based on bending dynamics, especially at the early stages of vesicle formation... To address how stochastic variation in the recruitment of CLCa-STAR and untagged CLCa and CLCb alters Δz measurements, we performed a Monte Carlo simulation of STAR measurements during CCV formation.”

My minor suggestions are:

Line 57: Flat lattices have been shown to contain pentagons (Sochacki et al. 2021). Possibly edit to “proposed by some models to contain mostly” or something similar.

Done @ line 63, and 67-68

Line 154: Please indicate here that pre-existing clathrin structures and longer-lived structure were excluded from the cohort analysis. The authors do discuss this later in the paper but I think this would focus the narrative of the work.

Done @ line 171-172

Line 199: Consider edit to “substantial clathrin accumulation” or similar for clarity.

Done @ line 217

Line 200: Consider edit to “robust accumulation” or similar for clarity.

Done @ line 218

Line 206: Consider edit to “unambiguously classify the site as a clathrin accumulation site” or similar for clarity.

Done @ line 224

Line 207: Consider edit of “identify” to “measure”.

Done @ line 225

Reviewer #3 (Remarks to the Author):

I'm overall satisfied with the responses to my reviews. I still think still that the authors could better flesh out what they mean by the flexible model of CME during their introduction.

Thank you. Done @ lines 74-79

“The flexible model encompasses CCM and FTC, as well as “in between” pathways. For example, 10% of total clathrin molecules could initially generate a flat structure, following the FTC model, and the remaining 90% of clathrin is recruited along with changes in curvature, following the CCM model. The flexible model remains open for many possible combinations of protein accumulation and membrane shape.”